# Tropical cyclone low-level wind speed, shear, and veer: sensitivity to the boundary layer parameterization in WRF

Sara Müller[1,2], Xiaoli Guo Larsén[1], and David Robert Verelst[1]

[1]Department of Wind and Energy Systems, Danish Technical University, Risø Lab/Campus Frederiksborgvej 399, Roskilde 4000

[2]Sino-Denmark Center (SDC) for Research and Education

**Correspondence:** Sara Müller (samul@dtu.dk)

**Abstract.** Mesoscale modeling can be used to analyze key parameters for wind turbine load assessment in a large variety of tropical cyclones. However, the modeled wind structure of tropical cyclones is known to be sensitive to the boundary layer scheme. We analyze modeled wind speed, shear, and wind veer across a wind turbine rotor plane in the eyewall and outer cyclone. We further assess the sensitivity of wind speed, shear, and veer to the boundary layer parameterization. Three model realizations of typhoon Megi over the open ocean using three frequently used boundary layer schemes in the Weather Research and Forecasting model are analyzed. All three typhoon simulations reasonably reproduce the cyclone track and structure. The boundary layer parametrization causes up to 15 % differences in median wind speed at hub height between the simulations. The simulated wind speed variability also depends on the boundary layer scheme. The modeled median wind shear is smaller or equal to 0.11 used in the current IEC standard regardless of the boundary layer scheme for the eyewall and outer cyclone region. However, up to 43.6 % of the simulated wind profiles in the eyewall region exceed 0.11. While the surface inflow angle is sensitive to the boundary layer scheme, wind veer in the lowest $400\,\mathrm{m}$ of the atmospheric boundary layer is less affected by the boundary layer scheme. Simulated median wind veer reaches values up to $1.7 \times 10^{-2}\,^\circ\mathrm{m}^{-1}$ ($1.2 \times 10^{-2}\,^\circ\mathrm{m}^{-1}$) in the eyewall region (outer cyclone region) and is relatively small compared to moderate wind speed regimes. On average, simulated wind speed shear and wind veer are highest in the eyewall region. Yet strong spatial organization of wind shear and veer along the rainbands may increase wind turbine loads, due to rapid changes in the wind profile at the turbine location.

## 1 Introduction

Offshore wind power has over the past decades become accessible for a wide region around the tropical and subtropical West Pacific. This region includes areas with large wind resources but it is frequently hit by tropical cyclones. For wind turbines in this region, tropical cyclones form the most extreme wind conditions, and therefore, the tropical cyclone wind field is a challenge for wind turbine design standards. Over the past decade turbine failures have been caused during different typhoons, such as Usagi, Rammasun, and Maria (Li et al., 2022). Tropical cyclones can cause fatigue failure of wind turbines (Chen et al., 2015; Chen and Xu, 2016; Chen, 2022). Further research incorporating mesoscale and microscale numerical models into aeroelastic wind turbine models is necessary to achieve reliable structural analysis of wind turbines in tropical cyclone conditions (Li et al., 2022).

To ensure the structural integrity of wind turbines, turbine design standards are defined in the International Electrotechnical Commission's standard (IEC) for onshore and offshore turbines (IEC, 2019a, b). The standards are based on site-specific wind speed classes and turbulence classes. The ability of wind turbines to withstand wind conditions within the turbine class is tested in design load cases (DLC). Different DLCs assess the loads acting on wind turbines during power production, and at stand-still. In and close to the eyewall wind speeds typically exceed the turbine-specific cutoff wind speed. In this case, turbines are parked to minimize loads. However, further away from the cyclone center, turbines may still be operating. Either way, the wind conditions tested in the DLCs consist of a mean wind profile combined with either a deterministic gust profile or turbulence. A power-law model is used for the wind profile with an associated hub height wind speed and wind shear over the rotor plane. A constant wind shear is suggested for load simulations of operating and parked turbines. Such a simplified wind shear model has an influence on turbine loading (Dimitrov et al., 2015). Measurements from He et al. (2016) suggest that the change in wind direction with height, wind veer, can be substantial in tropical cyclones. Similarly, Worsnop et al. (2017) find high gust factors, rapid directional changes, and substantial veer in tropical cyclones related to non-stationary small-scale structures in large-eddy simulations. Kapoor et al. (2020) show, that these features can substantially increase loads with respect to the cyclone-scale mean state. Both studies from Worsnop et al. (2017) and Kapoor et al. (2020) highlight that wind veer should be considered in wind turbine load assessment. Both studies are based on idealized Category 5 hurricane simulation. Sanchez Gomez et al. (2023) analyze wind shear and veer in idealized simulations of Category 1-3 Hurricanes. They find that both wind shear and wind veer exceed current design standards. Because of the high computational costs of large-eddy simulations, mesoscale simulations remain an attractive and important tool for assessing a large number of tropical cyclones with different intensities and storm sizes embedded in the large-scale circulation.

However, mesoscale models are bound to parameterize sub-grid-scale (SGS) turbulent transport of heat, momentum, and moisture at the sea surface and in the boundary layer. The relative size of these fluxes is crucial for the intensification of tropical cyclones (Emanuel, 1986). The impact of SGS turbulent fluxes on tropical cyclone simulations has been widely investigated. Ye et al. (2023a) show how the spatial distribution of SGS turbulent fluxes depends on the boundary layer closure and how the SGS fluxes affect the tropical cyclone wind field. It has been shown, that the choice of the boundary layer scheme affects the tropical cyclone intensity (Gopalakrishnan et al., 2013; Rai and Pattnaik, 2018; Rajeswari et al., 2020; Zhang et al., 2020; Ye et al., 2023b), the storm radius (Gopalakrishnan et al., 2013; Ye et al., 2023b), the boundary layer inflow strength (Gopalakrishnan et al., 2013; Rajeswari et al., 2020; Zhang et al., 2020) and the inflow layer depth (Rai and Pattnaik, 2018; Gopalakrishnan et al., 2013; Chen, 2022). This suggests, that for tropical cyclones, the three parameters, namely, mean wind speed, wind shear, and veer can be sensitive to the surface and boundary layer parametrization. For moderate wind conditions in mid-latitudes, simulated wind shear was found to be sensitive to the boundary layer parametrization and the model performance depends on atmospheric stability (Draxl et al., 2014; Krogsæter and Reuder, 2014). For tropical cyclone conditions, the sensitivity of wind shear and veer on the boundary layer parametrization might differ from these moderate wind regimes. It is important to know how large the associated uncertainty in these three parameters is. Because civil structures are within the lowest hundred meters it is especially important to focus on the lower part of the boundary layer.

Mesoscale atmospheric simulations use larger grid spacing than large-eddy simulations. Typically they can only resolve wind speed variability on scales on the order of seven times the horizontal grid spacing (Skamarock, 2004). Smaller scale structures resolved in LES simulations, such as roll vortices, cannot be adequately resolved in mesoscale simulations (Li et al., 2021). With that, the modeled spatial and temporal maximal values of a variable, such as wind speed, depend on the resolved model variability (Larsén et al., 2012). Yet, extreme values are important for structural design and load assessments, and maximal modeled wind speed is one of the most used for model verification (Rajeswari et al., 2020; Shenoy et al., 2021). Nolan et al. (2009b, a) show that the amount of high-frequency perturbations along the eyewall varies between mesoscale simulations with different boundary layer parameterizations. Zhu et al. (2014) analyze the mechanism leading to eyewall perturbations with different frequencies. They show, that the eyewall perturbations depend on vertical SGS fluxes. Similarly, Xu et al. (2021) show that SGS fluxes influence the fine-scale structure of the tropical cyclone wind field in the turbulent gray-zone. Given the effect of the boundary layer simulation on wind field perturbations, the wind speed variability is likely affected by the boundary layer parametrization.

Tropical cyclones have a characteristic wind field structure consisting of three regions. The tropical cyclone eye forms the storm center. There, wind speeds are low and wind turbine loads are expected to be small. The wind speeds are largest and often above wind turbine cut-out in the eyewall region. In the outer cyclone region winds are less extreme and wind turbines might still be operating. Wang et al. (2022) proposes a multi-stage framework to account for the difference in wind speed and turbulence profiles between the eyewall and rainband region.

Accordingly, we investigate the following aspects of the tropical cyclone wind field:

1. How much is the median wind speed, shear, and veer affected by the boundary layer scheme, and how do they compare to the IEC standard?

2. What is the distribution and variability of modeled wind shear, veer, and horizontal wind speed, and how does the distribution depend on the boundary layer parametrization?

3. How is wind speed, shear, and veer, spatially distributed and how does it differ between the eyewall and the outer cyclone region?

## 2 Methods

### 2.1 Model set up

The open-source Weather Research and Forecasting model (WRF) version 4.4 is used to simulate typhoon Megi, which hit Taiwan in September 2016. This case is chosen for a couple of reasons. First of all, Megi is one of the most severe storms that affected the region over Taiwan. As such it serves as a good example of severe wind conditions. Second, Synthetic Aperture Radar (SAR) data are available, in addition to the best track data sets (described in Sect. 2.3).

We compare three simulations using three different boundary-layer parametrization schemes, summarized in Table 1. The purpose is to evaluate the spread between the best physics suits for the different boundary layer schemes. Therefore, each boundary layer scheme is combined with the surface layer scheme that it has been developed with.

1. The Mellor-Yamada-Janjic boundary layer scheme (MYJ) (Janić, 2001) with the revised Eta similarity surface layer scheme (Janić, 2001).

2. The Mellor-Yamada-Nakanishi-Niino order 2.5 boundary layer scheme (MYNN) (Nakanishi and Niino, 2009) with the MYNN surface layer scheme (Nakanishi and Niino, 2009).

3. The Yonsei University boundary layer scheme (YSU) (Hong et al., 2006) with the revised MM5 surface layer scheme (Jiménez et al., 2012).

The three schemes use different ways to calculate SGS turbulent fluxes. The YSU scheme uses a non-local first-order K-closure. The MYJ and MYNN schemes use a 1.5 order local Turbulent kinetic energy (TKE) closure. MYNN is formulated based on variables conserved for moist reversible adiabatic processes and is therefore often called a "moist" scheme (Zhu et al., 2014). Differently, YSU and MYJ are "dry" schemes. The three schemes are widely used in WRF. Due to its non-local closure, YSU is a popular choice to simulate tropical cyclones. Many studies analyze tropical cyclones simulated with the MYJ scheme (Nolan et al., 2009b; Sparks et al., 2019; Rajeswari et al., 2020; Shenoy et al., 2021), partly because it was one of only two boundary layer options in the earlier version of WRF (V2.2). The MYNN scheme is an important option for wind resource assessment in the presence of wind farm effects, because wind turbine parametrizations are available for the scheme (Fitch et al., 2012; Volker et al., 2015).

The high wind speeds in tropical cyclones affect the surface fluxes. Under moderate wind conditions, the surface momentum flux over water is typically modeled using the Charnock relation (Charnock, 1955). In the Charnock relation, the drag coefficient ($C_d$) increases monotonically with the wind speed. As a default, the MYJ and the MYNN scheme are based on the Charnock relation and feature monotonically increasing $C_d$ with wind speed. However, observations in Powell et al. (2003) and Donelan et al. (2004) suggest that $C_d$ levels off for wind speeds larger than $33 \mathrm{~m~s}^{-1}$. Correcting the drag coefficient towards these findings was shown to improve modeling results (Nolan et al., 2009b). In WRF version 4.4, ETA similarity surface layer scheme is the only surface layer scheme with the option to account for such a dependence of $C_d$ on wind speed. In WRF version 4.4, it can be selected over the *isftcflx* option. We use the *isftcflx* option 2 for the YSU simulation. This option has a constant drag coefficient for wind speeds faster than $33 \mathrm{~m~s}^{-1}$ (Green and Zhang, 2013). In the default, the exchange coefficients for sensible and latent heat are a function of $C_d$. Using the *isftcflx* option 2, these coefficients are modified based on Garratt (1994). At $40 \mathrm{~m~s}^{-1}$ surface wind speeds, the exchange coefficients of sensible and latent heat are 40 % to 50 % smaller than in the default (Green and Zhang, 2013).

In all simulations, the Thompson scheme is used to parameterize the micro-physical processes (Thompson et al., 2008). On the outermost domain, convective clouds are parametrized by the Kain-Fritch (Kain, 2004) cumulus scheme. The longwave

**Table 1.** Summary of parametrization schemes used in the three simulations with YSU, MYJ, and MYNN 2.5.

| Name | YSU | MYJ | MYNN |
|---|---|---|---|
| Boundary layer scheme | YSU | MYJ | MYNN 2.5 |
| Surface layer scheme | MM5 | Eta similarity | MYNN surface layer |
| Ocean surface drag | isftcflx=2 | unchanged | unchanged |
| Micro physics scheme | | Thomson et al. | |
| Cumulus scheme in d01 | | Kain-Fritsch | |
| Radiation physics scheme | | RRTMG | |

and shortwave radiation is parametrized by the RRTMG (Iacono et al., 2008) scheme. Even though offshore wind projects are mostly limited to coastal regions, we chose to focus on the cyclone intensification stage over open water before the typhoon makes landfall. This stage is chosen for two reasons: 1.) Temporal and spatial averaging of the wind field is only reasonably applicable in the absence of abrupt surface changes. 2.) The comparison to literature is simplified, as the majority of model studies addressing the tropical cyclone wind structure focus on tropical cyclones over the open ocean. How the wind field over land and in close proximity to land differs from the open ocean should be addressed in further studies, where our study can serve as a baseline. The domain setup is displayed in Fig. 1. WRF is run on three one-way nested domains, where the two innermost domains use the vortex following grid configuration. All three domains are initialized at 00 UTC on 25 September. The first 12 h are used as spin-up time. The following 36-hour simulations starting at 12 UTC on 25 September are used for the analysis. The outer domain has $350 \times 361$ grid points with a horizontal grid spacing of 18 km. The two inner domains have $361 \times 361$ grid points and a horizontal spacing of 6 km and 2 km. The three domains are run with a 45, 15, and 5 s timestep. All domains use 70 vertical layers. The lowest model levels have a mean height of 8, 26, 47, 72, 102, 139, 183, 234, 297, and 372 m. The model top is at 200 hPa. For the initial and boundary conditions, ERA5 reanalysis data (Hersbach et al., 2018) is used. The sea surface temperature is used from OSTIA (Donlon et al., 2012).

## 2.2 Analysis method

Wind speed, wind shear, and veer are calculated from the three simulations. The analysis is based on the instantaneous model output of the innermost domain, saved every ten minutes. The analysis of wind shear is performed, such that a comparison to the IEC standard is straightforward. For the assessment of extreme wind conditions, the IEC assumes a wind profile according to the extreme wind speed model:

$$V_{50} = V_{ref} * (\frac{z}{z_{hub}})^\alpha \tag{1}$$

Here $V_{50}$ is the extreme wind speed with a return period of 50 years, averaged over a ten minute interval. For areas affected by tropical cyclones, the reference wind speed $V_{ref}$ is 57 $\mathrm{m\,s^{-1}}$. The height is given by $z$, and the hub height by $z_{hub}$. The

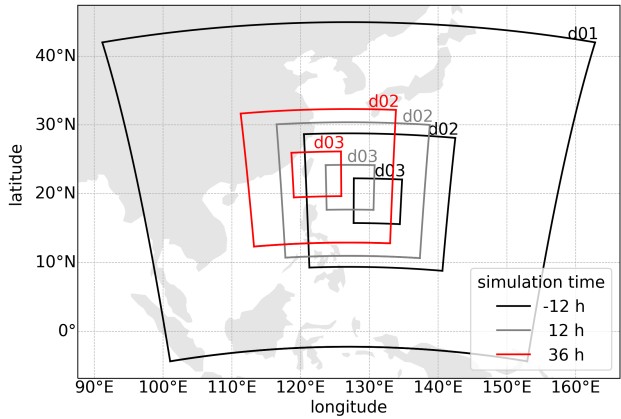

**Figure 1.** Domain set up: domain borders at the start of the 12 h spin-up time (black), after 12 simulation hours (gray), and after 36 simulation hours (red). Note that domain d01 is fixed in time.

wind shear exponent $\alpha$ is a measure of vertical wind shear. Strong wind shear is associated with larger $\alpha$ values. For extreme wind conditions, the IEC uses a constant $\alpha$ of 0.11 (IEC, 2019a). Under normal wind conditions offshore, $\alpha$ is set to 0.14 (IEC, 2019b). In this study $\alpha$ is calculated using Eq. 2.

$$\alpha = \frac{\ln(u_2/u_1)}{\ln(z_2/z_1)} \tag{2}$$

Here, $u_1$ and $u_2$ are the wind speeds at heights $z_1$ and $z_2$ respectively. While $\alpha$ depends on height, the IEC assumes a constant $\alpha$ over the rotor plane. We analyze both $\alpha$ between consecutive model levels at different heights and the total $\alpha$ over the rotor plane. For the latter, we use a least square fit between $\ln(u)$ and $\ln(z)$. All model levels between the rotor bottom and the rotor top are used for the fit.

For the rotor bottom, we use the height of the second model level, which has a median height of 26 m. The eighth model
level with a median height of 234 m is used as the height of the rotor top. The heights are chosen as a compromise between using model-level heights and representing a wide range of future wind turbine types planned in the Taiwan Strait based on 4Coffshore (2023). The described fictional turbine has a rotor diameter of 208 m and a hub height of 130 m. While most operating turbines in the Taiwan Strait have smaller rotor diameters, 14 MW wind turbines with 222 m rotor diameter are now planned in the Hailong Offshore wind farm to be operational in 2026. We further analyze the wind speed, at the fifth model
level. This model level has a medium height of 139 m and is closest to the hub height of the fictional wind turbine.

Wind veer is defined as the shortest rotational path between the wind direction (WD) at different heights, which is maximal 180 °. It is defined such that positive (negative) veer values describe clockwise (counterclockwise) rotation with increasing height. With that, a decreasing (increasing) inflow angle with height is associated with positive (negative) veer values. In this study we normalize wind veer by the vertical distance between two used heights. Veer is calculated as follows:

$$\text{Veer} = \frac{\min(\text{WD}_2 - \text{WD}_1)}{z_2 - z_1} \tag{3}$$

Here, WD is defined at the model heights $z_1$ and $z_2$. Similar to the analysis of $\alpha$, wind veer is analyzed between consecutive model levels at different heights, as well as over the rotor plane. For the latter, we use a least square fit between WD and $z$. All model levels between the rotor bottom and the rotor top are used for the fit.

The eyewall and outer cyclone regions are analyzed separately. With that, we can account for and characterize differences between the two storm regions. To avoid, that the position of the simulation domain relative to the cyclone center influences the analysis, we use only grid points within a distance to the cyclone center (R) smaller than 350 km. The definitions of the eyewall and outer region are illustrated in Fig. 2 and summarized in Eq. 4 and 5. We define the eyewall as a high wind speed regime. Our definition is based on the simulated wind speed at 10 m ($WS_{10}$) at each simulation time step. Grid points are assigned to the eyewall region if two criteria are fulfilled: 1.) $WS_{10}$ is greater than or equal to the 80th percentile of $WS_{10}$ ($P_{80}(WS_{10})$), and 2.) R is less than 250 km. The selected eyewall area is 76000 km$^2$. This definition of the eyewall region includes high wind speed areas of the inner rainbands and potentially outer rainbands. Therefore the eyewall region should not be interpreted as the narrow eyewall in tropical cyclones, but rather as an extended high wind speed area. The thickness of the eyewall is not symmetrical over the azimuth and can be zero. Note also, that the area includes gaps, where the wind speed is lower than $P_{80}(WS_{10})$. The outer region includes all grid points that are not part of the eyewall and the eye. To distinguish between the outer region and the eye, we define a critical radius $R_{Eye}$ in Eq. 5. Here, $R_{Eye}$ is taken as the 10th percentile of R of the eyewall grid points ($P_{10}(R_{\text{Eyewall}})$). Thus

$$
\text{Region} = \begin{cases} \text{Eyewall:} & \text{if } WS_{10} \geq P_{80}(WS_{10}) \text{ and } R < 250\,\text{km}, \\ \text{Outer cyclone:} & \text{if not Eyewall and } R > R_{Eye} \end{cases} \tag{4}
$$

where

$$
R_{Eye} = P_{10}(R_{Eyewall}) \tag{5}
$$

The outer region is not homogeneous. It includes both the rainbands and the non-convective moat areas. These areas have different properties. However, we do not separate them, to prevent the analysis from becoming sensitive to the selection criteria.

Radial averages, profiles, and probability density distributions are calculated using all 144 output timesteps from 12 UTC on 25 September to 12 UTC on 26 September. Later timesteps are not included, because of enhanced wind field asymmetries near and over Taiwan. We note that typhoon-land interactions are clearly important for wind turbines, and they are addressed in Müller et al. (2023). All grid points within the defined radius range are used, resulting in over 2.7 million points for the eyewall region and 10.4 million in the outer cyclone region. The median and the interquartile range (IQR) are compared between the simulations.

In this study, the cyclone track, inflow angle, and differentiation between the outer cyclone and eyewall region are based on the definition of the cyclone center. We obtain the cyclone center with an algorithm based on the minimal variance of the sea level pressure (SLP) over bands with equal distance to the cyclone center. This center detecting method is recommended by Yang et al. (2020), because it leads to a smooth track variation over time and enhanced symmetry in the wind field.

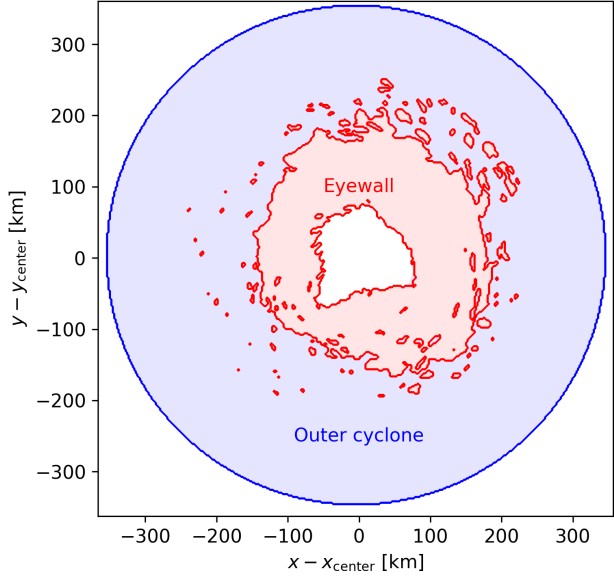

**Figure 2.** Eyewall and outer cyclone region at an example timestep. The x and y axes show the distance from the cyclone center.

The one-dimensional power spectrum in the wave number domain is used as a measure of variability in the horizontal 10 m wind speed as in Skamarock (2004). For each time step one-dimensional spectra are calculated over the model domain for each model row (oriented approximately in the west-east direction) and column (oriented approximately in the north-south direction). Before calculating the spectra, linear trends in the rows and columns are removed by individually subtracting the result of a linear least-squares fit. The spectrum is obtained by Fourier transform. Ten lines of grid points are removed from the model domain edges to avoid the sponge-layer effects related to the domain nesting. This results in $2 \times 341$ spectra per time step. These spectra are averaged over all 144-time steps between 12 UTC on 25 September and 12 UTC on 26 September. The purpose of the spectral analysis of the wind field is to examine the resolved wind variability in comparison with theory as well as with SAR wind data.

## 2.3 Validation method

The model data is compared to and qualitatively validated against the best track data. Best track data is publicly available from different meteorological centers. In this study best track data sets from two centers are used: the US Joint Typhoon Warning Center (JTWC) and the Regional Specialized Meteorological Center (RSMC) Tokyo-Typhoon Center operated by the Japan Meteorological Agency (JMA). Both data sets include the cyclone's central position in three to six-hour intervals. The best track data sets further include the central pressure and the maximal sustained wind speed. These are mainly based on the method described by Dvorak (1984) (RSMC, 2021; Chu et al., 2002). The maximal sustained wind speed is defined differently in the two data sets. JTWC reports the maximal one-minute sustained wind speed (Chu et al., 2002), defined as the

maximal 10 m wind speed averaged over 1 minute encountered over the entire cyclone structure. JMA in contrast reports the maximal ten-minute sustained wind speed, giving the maximal 10 minute average wind speed (RSMC, 2021). The conversion between the two metrics is not straightforward. Harper et al. (2010) recommends a conversion factor of 0.93 between the larger one-minute sustained wind speed and the ten-minute sustained wind speed over the ocean. Chu et al. (2002) states that the one-minute sustained wind speed is in general around 14 % larger than the ten-minute sustained wind speed. This results in a conversion factor of 0.88, and a larger difference between the two data sets. The difference between the sustained wind speed in the two data sets is even larger than 14 % (Ott, 2006). We decided to use both best track data sets to see the model spread in relation to the spread in the best track data. For easier comparison, we additionally provide the JTWC one-minute sustained wind speeds converted to ten-minute sustained wind speeds. We use, the factor 0.93 recommended by Harper et al. (2010) for the conversion.

We further use Synthetic Aperture Radar (SAR) wind scenes for the validation of the modeled horizontal wind speed structure and variability. Wind scenes are post-processed by Badger et al. (2022) and taken from the European Space Agency (ESA). This study uses eight wind scenes, that cover different areas of typhoon Megi between 9 and 22 UTC on 26 September 2016. The scenes are shown in Fig. 3. The scenes provide the 10 m wind speed in a regular 500 m grid. One-dimensional power spectra are calculated from the eight wind scenes in the same way as for the model data (see Sect. 2.2). The spectra are calculated over the axis with the larger number of grid points and averaged over the shorter axis for each of the eight SAR wind scenes.

Lastly, the wind profile structure is compared to global positioning system dropsonde measurements documented in Powell et al. (2003) and Vickery et al. (2009). The measurements include 331 profiles from 15 tropical cyclones over the Atlantic Eastern and Central Pacific. Based on these measurements Vickery et al. (2009) suggests an empirical formulation for the tropical cyclone boundary layer which accounts for a low-level jet:

$$U(z) = u_*/\kappa [ln(z/z_0) - a(z/H^*)^n], \tag{6}$$

Here, $u_*$ is the friction velocity, $\kappa$ the von Kármán coefficient, $z$ the height, $z_0$ the surface roughness length, and $H^*$ a boundary layer height parameter. The parameters $a$ and $n$ are free parameters fitted to the dropsonde measurements. Vickery et al. (2009) analyze the dropsonde measurements in a composites sense. They group the measurements according to the radius of maximal winds (RMW) and the mean boundary layer (MBL) wind speed. The latter is defined as the mean wind speed over a height range of 10 to 500 m. Based on the JTWC best track data set, typhoon Megi's RMW is mostly in the range of 30-60 km during the analyzed period. The MBL wind speed is calculated from the simulated eyewall profiles. Depending on the boundary layer scheme used in the simulation, the MBL wind speed is in the range of 30-39 $\mathrm{m\,s^{-1}}$ or 40-49 $\mathrm{m\,s^{-1}}$. With that, we can compare the simulated wind profiles with wind profiles defined by parameter-sets given by Vickery et al. (2009), describing wind profiles with the corresponding RMW and the two MBL wind speed ranges. Vickery et al. (2009) further assess two methods to obtain the parameter-sets for each group. Both parameter-sets are used in our study. To compare the vertical wind shear from the dropsonde measurements to the analyzed simulations, $\alpha$ is calculated from Eq. 2.

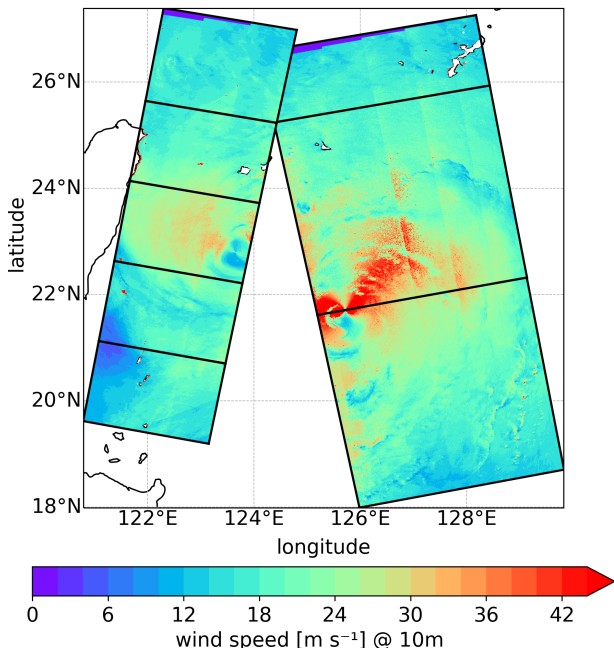

**Figure 3.** Synthetic aperture radar wind scenes used for model validation. The five wind scenes on the western side are taken between 9.35 and 9.38 UTC and the three on the eastern side between 21.31 and 21.46 UTC on 26 September 2016. The wind scenes are retrieved from Badger et al. (2022).

## 3 Results

### 3.1 Model verification against best track data

Cyclone Megi develops from a tropical disturbance in the western Pacific Ocean and reaches tropical cyclone intensity on 24 September. Megi's track and intensity in terms of minimal SLP and maximal wind speed from 25 September onward are shown in Fig. 4 and 5. On 24 September cyclone Megi continues its trajectory north-westwards toward Taiwan. During this trajectory over the open ocean, Megi intensifies. Its minimal SLP decreases and the maximal wind speed increases and reaches a maximum at 00 UTC on 27 September (see Fig. 5). On 27 September Megi hits Taiwan and weakens. This can be seen in the consequent increase of the minimal SLP and the decrease in wind speed. After entering the Taiwan Strait, Megi makes landfall over mainland China between 18 UTC on 27 September and 00 UTC on 28 September.

The simulations cover the period between 12 UTC on 25 to 00 UTC on 27 September when Megi intensifies over the open ocean. During this time, the two best track data sets are in close agreement in terms of central position (Fig. 4). All three simulations can reproduce the general cyclone track, with only a slight southward deflection. The error at the end of the simulations is within 130 km. The error in the simulated track is larger than in the ERA5 data set, which is used as boundary conditions. The simulations can further reproduce cyclone intensification (Fig. 5). However, the degree of intensification varies

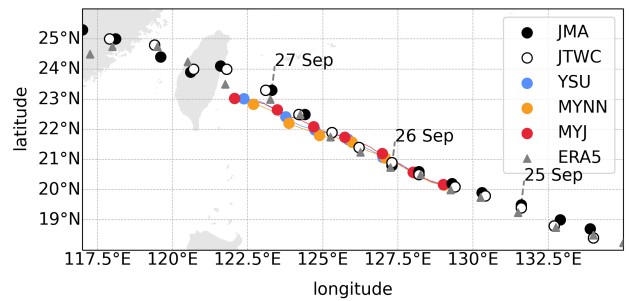

**Figure 4.** Typhoon track: Cyclone center position of the YSU (blue), MYNN (red), MYNN-eta (purple), and MYJ (yellow) simulation and of the best track data sets from JMA (black filled points) and JTWC (white filled points). Points show the position at 00, 06, 12, and 18 UTC. Lines show the position every 10 minutes for the simulations.

between the simulations. From Fig. 5 it can be seen that the minimal SLP drops initially at the highest rate in the MYJ simulation. However, at 6 UTC on 26 September, the MYJ simulation stops intensifying. At the simulation end, the minimal SLP of the MYJ and the YSU schemes are similar. The comparison between the maximal wind speed and minimal SLP in the simulations and the best track data has shortcomings, especially since the two sizes depend on the spatial and temporal resolution of the simulation. Nevertheless, the comparison is widely used and helpful to qualitatively evaluate the simulated intensity (Rajeswari et al., 2020; Shenoy et al., 2021; Zhu et al., 2014). The minimal SLP of both MYJ and YSU is mostly between the minimal SLP from the JTWC and the JMA best track data set. The difference in the simulated minimal SLP between these two simulations is smaller than between the best track data sets. Similarly, the maximal wind speed of the YSU and the MYJ simulation is between the maximal ten-minute sustained wind speed reported by JMA and the maximal one-minute sustained wind speed reported by JTWC. Differently, the typhoon in the MYNN simulation intensifies less than in the JMA and JTWC best track data sets. Its minimal SLP follows the higher minimal SLP of the coarser ERA5 data. The MYNN maximal wind speed follows the maximal ten-minute sustained wind speed reported by JMA and is lower than in the YSU and MYJ simulations.

## 3.2 Mean wind field

We analyze the characteristic structure of a tropical cyclone through an example simulation timestep at 00 UTC on 26 September. This timestep is the center of the analyzed period from 12 UTC on 25 September to 12 UTC on 26 September. The horizontal wind field at $10\,\mathrm{m}$ at that time is given in Fig. 6 for the three different boundary layer schemes. In the eye, the center of the storm, wind speeds are near zero. Outside of the eye, the wind rotates in a circular pattern counterclockwise. Larger wind speeds are evident in the back right quadrant, northeastward of the eye. Here, wind speeds are around $4\,\mathrm{m\,s^{-1}}$ larger than in the back left quadrant, southwestward of the eye. Nevertheless, the wind speed changes similarly with increasing distance to the cyclone center in all quadrants. This allows averaging the wind field over the azimuth, as shown in Fig. 7. Based on the averaged wind over time and azimuth, we can systematically compare the wind speed in the three simulations. Wind speeds are

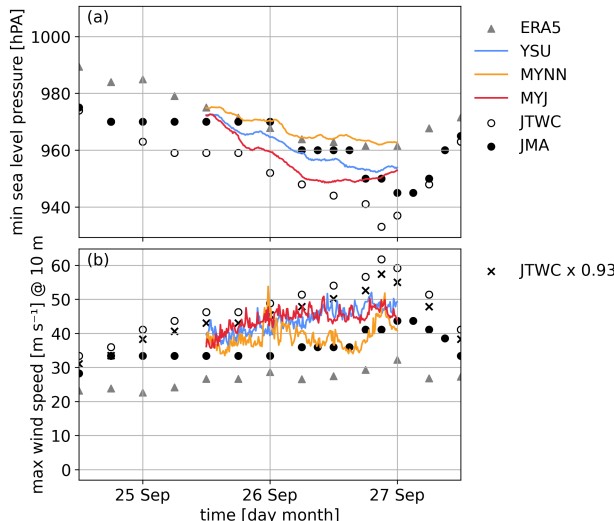

**Figure 5.** Cyclone intensity in terms of a) minimal SLP of the simulations (lines) and the ERA5 reanalysis data (triangles) compared to the best track data (circles) and b) maximal instantaneous wind speed at 10 m in the simulations (lines), the ERA5 reanalysis (triangles), the maximal ten-minutes sustained wind field from JMA (black points), and the maximal one-minute sustained wind field from JTWC (white points). Black crosses show the JTWC values multiplied by 0.93, as recommended by Harper et al. (2010) to convert one-minute sustained wind speeds to ten-minute sustained wind speeds in tropical cyclones over the sea.

maximal in the eyewall. In the YSU and MYJ simulations, the wind speed at 10 m is $32\,\mathrm{m\,s}^{-1}$ in the eyewall. The MYNN simulation shows a $12\,\mathrm{m\,s}^{-1}$ lower 10 m eyewall wind speed than the MYJ and the YSU simulation. This qualitatively agrees with the lower maximal wind speed in the MYNN simulation over the entire simulation period as described in Sect. 3.1. Addition-

ally, the distance between the eye and the maximal wind speed in the eyewall is larger in the MYNN (104 $\mathrm{km}$) simulation than in the YSU (92 km) and the MYJ (94 km) simulations. With increasing distance from the eyewall outwards, the wind speed gradually decreases. The radial gradient in wind speed is most pronounced in the MYJ simulation. In the outer cyclone region, the surface wind speeds are highest in the YSU simulation, followed by the MYJ simulation and the MYNN simulation.

     The simulated vertical wind field structure is analyzed based on median profiles. The wind speed increases with height as

shown in Fig. 8. The structure of the simulated profiles qualitatively agrees with the structure of dropsonde measurements reported by Vickery et al. (2009). The simulated profiles are characterized by a jet at around 800 m in the eyewall region and at 1200 m in the outer cyclone region. Below the jet, the simulations have an approximately logarithmic wind speed increase with height. The simulated wind profiles differ in two aspects: Firstly, the wind shear below the jet nose is more pronounced in the YSU simulation with respect to the profiles from Vickery et al. (2009) and the MYNN and MYJ simulations. Secondly,

the slope of the logarithmic wind profile is larger in the MYJ simulation, than in the profiles from Vickery et al. (2009), the MYNN simulation, and particularly in the YSU simulation.

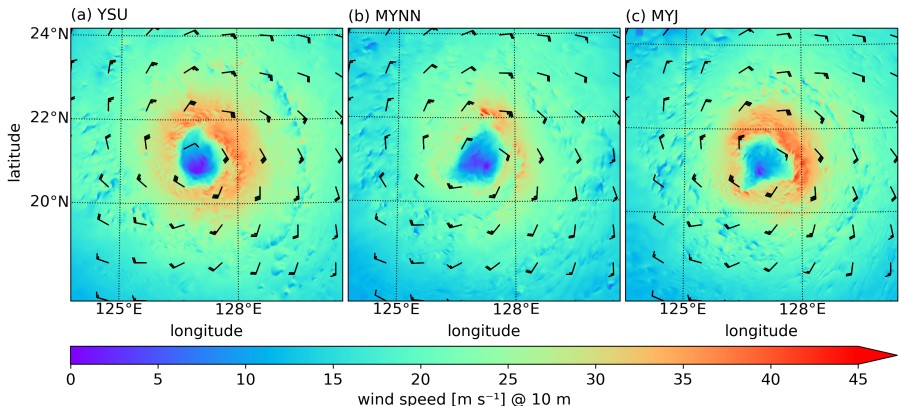

**Figure 6.** Wind speed at 10 m taken from the model output at 00 UTC on 26 September for the a) YSU, b) MYNN, and c) MYJ simulation.

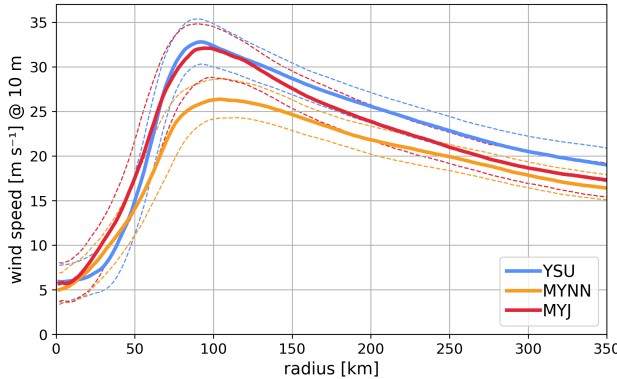

**Figure 7.** 10 m wind speed as a function of distance from the cyclone center for the YSU, MYNN, and MYJ simulation: median (solid line) and 0.25 and 0.75 percentiles (dashed lines). The values are obtained from all grid points within 144 output time steps between 12 UTC on 25 September and 12 UTC on 26 September, analyzed in bins of 2 km radius.

To evaluate the change in wind speed with height Table 2 lists the median $\alpha$ between rotor top and rotor bottom. The simulated median $\alpha$ ranges from $8.3 \times 10^{-2}$ to $1.1 \times 10^{-1}$ in the eyewall. This is in good agreement with the $\alpha$ values obtained from the profiles form Vickery et al. (2009), which are within $9.1 \times 10^{-2}$ and $9.6 \times 10^{-2}$ for the selected profiles. Compared to the eyewall region, the simulated $\alpha$ values are smaller in the outer cyclone region, where they are in the range of $7.0 \times 10^{-2}$ to $9.8 \times 10^{-2}$. The larger slope in the MYJ wind speed profile with respect to YSU reflects in a $2.7 \times 10^{-2}$ to $2.8 \times 10^{-2}$ larger median shear exponent. While the wind profile in the IEC standard is based on a constant $\alpha$ over the rotor diameter, this simplification might not be given for large turbine sizes. To analyze how $\alpha$ varies with height, profiles of $\alpha$ are shown in panels c and d of Fig. 8. According to Vickery's wind speed model (Eq. 6), $\alpha$ decreases monotonically with height. This is different in the simulations. The YSU and MYJ simulations show a similar behavior of $\alpha$ with height. In these simulations, $\alpha$ decreases with height only below the hub height. Between the hub height and the rotor top the median $\alpha$ changes less with

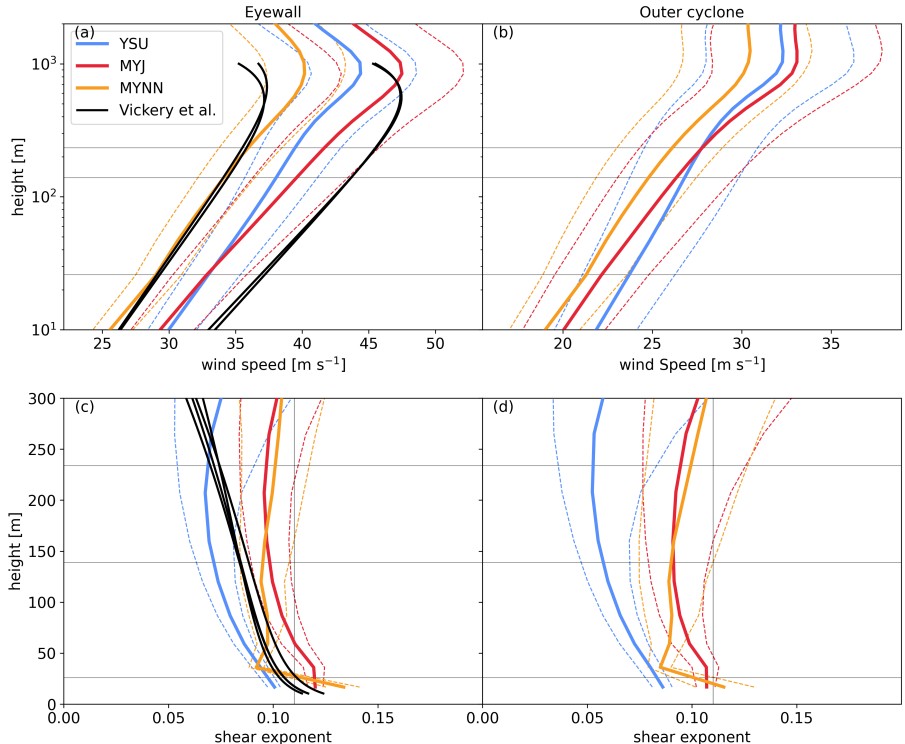

**Figure 8.** a, c) and b, d) show the vertical profiles of wind speed and shear exponent for the eyewall region and outer cyclone, respectively. Solid lines show the median and dashed lines show the 0.25 and 0.75 percentiles. Horizontal lines show the heights of 26, 139, and $234\,\mathrm{m}$. A shear exponent of 0.11 is indicated by vertical lines. The values are obtained from all grid columns within defined regions in 144 output time steps, between 12 UTC on 25 September and 12 UTC on 26 September.

height. Yet, along the spiraling rainbands, $\alpha$ increases around $3 \times 10^{-2}$ between the hub height and the rotor top (not shown). The MYNN simulation produces a pronounced change in wind speed between the first and second model levels. This results in an enhanced $\alpha$ between these levels. Above the second model level $\alpha$ increases slightly. Over the rotor diameter the increase

in $\alpha$ is $7 \times 10^{-3}$ ($2 \times 10^{-3}$) in the outer cyclone (eyewall) region.

The difference in the slope of the wind profile is important for wind turbines because it controls the wind speed at hub height and the wind shear over the rotor plane. At hub height, the horizontal wind speed has an analog structure as the surface wind field but an increased magnitude. The MYJ simulation has a $1.9\,\mathrm{m\,s^{-1}}$ larger eyewall wind speed at hub height than the YSU simulation (see Table 2). Note, that this is different at 10 m, where the two simulations have similar wind speeds in the eyewall.

The difference between the two heights is a direct result of the larger shear exponent in the MYJ simulation with respect to the YSU simulation. At hub height, the MYNN simulation has a $7.9\,\mathrm{m\,s^{-1}}$ ($6.0\,\mathrm{m\,s^{-1}}$) smaller wind speed than MYJ (YSU). The wind speed in the outer cyclone region at hub height is similar in the MYJ and the YSU and smaller in the MYNN simulation.

**Table 2.** Median and interquartile range (IQR) of wind speed at $139\,\mathrm{m}$, wind shear exponent, and wind veer, as well as the percentage of shear exponent values larger than 0.11. The values are listed for the eyewall region and outer cyclone region for the YSU, MYJ, and MYNN simulation.

| Region | Scheme | Wind speed [$\mathrm{m\,s^{-1}}$] | | Shear exponent | | | Wind veer [$^{\circ}\,\mathrm{m^{-1}}$] | |
|---|---|---|---|---|---|---|---|---|
| | | median | IQR | median | IQR | % > 0.11 | median | IQR |
| | YSU | 38.0 | 5.4 | $8.3 \times 10^{-2}$ | $1.1 \times 10^{-2}$ | 0.9 | $1.4 \times 10^{-2}$ | $7.8 \times 10^{-3}$ |
| Eyewall | MYNN | 34.1 | 4.3 | $9.6 \times 10^{-2}$ | $1.2 \times 10^{-2}$ | 6.5 | $1.6 \times 10^{-2}$ | $7.1 \times 10^{-3}$ |
| | MYJ | 39.5 | 7.1 | $1.1 \times 10^{-1}$ | $1.3 \times 10^{-2}$ | 43.6 | $1.7 \times 10^{-2}$ | $7.6 \times 10^{-3}$ |
| | YSU | 26.8 | 6.1 | $7.0 \times 10^{-2}$ | $1.6 \times 10^{-2}$ | 3.8 | $9.1 \times 10^{-3}$ | $8.0 \times 10^{-3}$ |
| Outer cyclone | MYNN | 24.8 | 5.6 | $8.9 \times 10^{-2}$ | $1.8 \times 10^{-2}$ | 10.4 | $1.2 \times 10^{-2}$ | $7.8 \times 10^{-3}$ |
| | MYJ | 26.3 | 6.5 | $9.8 \times 10^{-2}$ | $1.9 \times 10^{-2}$ | 22.3 | $1.2 \times 10^{-2}$ | $8.4 \times 10^{-3}$ |

The inflow angle is shown as a function of height in Fig. 9. All schemes exhibit a median inflow with a depth of around $1000\,\mathrm{m}$. The surface inflow angle is smaller in the MYNN simulation than in the YSU and the MYJ simulations. Within the inflow layer, the mean wind turns outward with respect to the cyclone center with height. The change in the inflow angle with height is relatively constant in the lowest $400\,\mathrm{m}$ of the boundary layer and comparable between the three simulations. The resulting median wind veer is in close agreement between the three simulations (see Table 2). The simulated median wind veer ranges from $1.4 \times 10^{-2}\,^{\circ}\,\mathrm{m^{-1}}$ to $1.7 \times 10^{-2}\,^{\circ}\,\mathrm{m^{-1}}$ in the eyewall region and from $9.1 \times 10^{-3}\,^{\circ}\,\mathrm{m^{-1}}$ to $1.2 \times 10^{-2}\,^{\circ}\,\mathrm{m^{-1}}$ in the outer cyclone region. Similar to our analysis of $\alpha$, we analyze the change in wind veer over the rotor diameter in Fig. 9 c and d. Wind veer is maximal close to the surface. Below the hub height veer generally decreases with height. The difference in wind veer between rotor bottom and hub height is 18 % in the MYJ simulation and around 8 % in the MYNN and YSU simulations. Above the hub height wind veer is nearly constant for the MYJ and slightly increases with height for the YSU and MYNN simulations. MYNN produces strongly enhanced wind veer between the lowest two model levels.

### 3.3 Wind variability

Apart from the typhoon-scale structure, the simulations also produce mesoscale variability in the wind field. Mesoscale wind fluctuations can be seen in all simulations within and outside of the eyewall in Fig. 6. Variability caused by turbulence is however not resolved in the mesoscale simulations. To address the resolved variability of wind speed, wind shear, and wind veer, we show the probability density functions of these sizes in Fig. 10. In Sect. 3.2 differences in median values were commented. Here, we focus on the spread of the distributions. The hub height eyewall wind speed ranges between 30 to $60\,\mathrm{m\,s^{-1}}$ (28 to $50\,\mathrm{m\,s^{-1}}$) in the MYJ and YSU (MYNN) simulation. Note, that the lower boundary of the distribution is controlled by our definition of the eyewall region (see Sect. 2). The wind speed distribution is broader for the MYJ simulation than for the YSU and particularly the MYNN simulation. Concretely, the IQR of the eyewall wind speed is $7.1\,\mathrm{m\,s^{-1}}$ in the

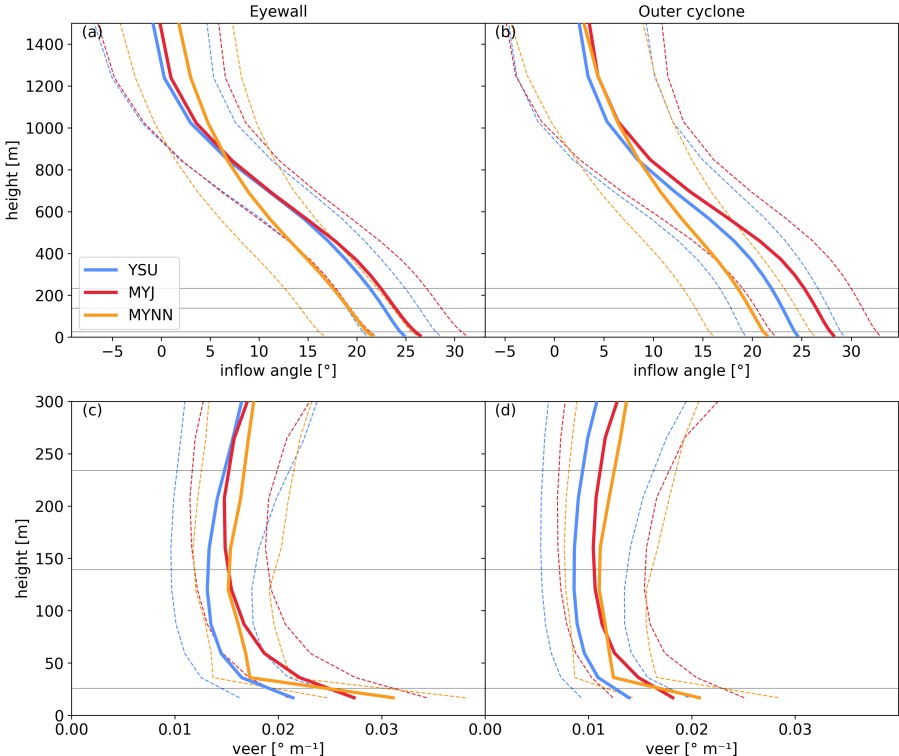

**Figure 9.** a, c) and b, d) are the vertical profiles of inflow angle and wind veer for the eyewall region and outer cyclone, respectively. Solid lines show the median and dashed lines show the 0.25 and 0.75 percentiles. Horizontal lines show the heights of 26, 139, and $234\,\text{m}$. The values are obtained from all grid columns within defined regions in 144 output time steps, between 12 UTC on 25 September and 12 UTC on 26 September.

MYJ simulation, $5.4\,\text{m}\,\text{s}^{-1}$ in YSU, and $4.3\,\text{m}\,\text{s}^{-1}$ in MYJ (see Table 2). Similarly, the wind speed distribution is broader in the outer cyclone region for the MYJ simulation, than for the other two simulations.

To investigate which scales contribute to the larger wind speed variability in the MYJ simulation, the one-dimensional wind speed power spectra in the wave number domain are analyzed in Fig. 11. In agreement with the wider wind speed distribution, MYJ shows the highest power spectral density and MYNN the lowest over the calculated wavelength range of 6 to $300\,\text{km}$.

The spectra can be divided into two parts:

1. For wavelengths larger than $15\,\text{km}$, the YSU and MYNN simulated spectra have a slope of approximately minus five-thirds and a slightly smaller slope for MYJ. The smaller slope in the MYJ simulation particularly increases the spectral power density contribution from higher wave numbers. The MYJ slope in this range fits the mean spectral slope obtained from the SAR images the best. Also, the magnitude shows the best fit for the MYJ simulation.

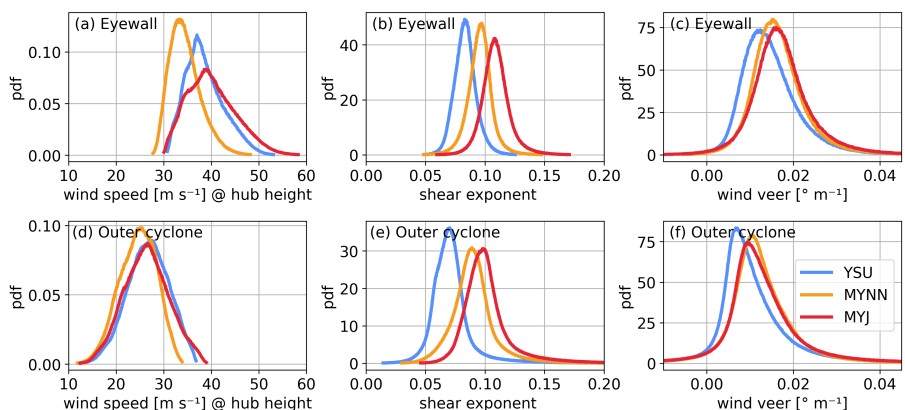

**Figure 10.** Probability density of a,d) wind speed at 139 m, b,e) wind shear exponent, and c,f) wind veer. The probability density is given for the a-c) eyewall region, and c-f) outer cyclone region for the YSU, MYJ, and MYNN simulation.

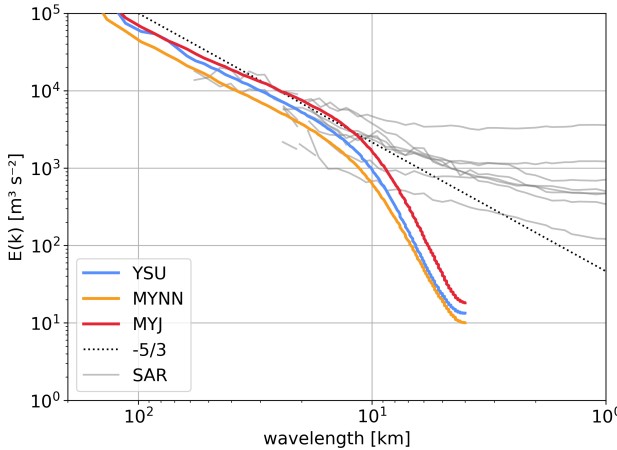

**Figure 11.** One-dimensional power spectrum in the wave number domain for the three simulations (color) and eight SAR wind scenes (gray). The slope of minus five-thirds is shown for reference (dashed). For better readability, logarithmic averaging is applied to the spectra.

2. For wavelengths smaller than 15 km the simulated spectra have a steeper slope than minus five-thirds. Differently, the spectra from the SAR images have flatter slopes than minus five-thirds. In other words, the energy level of the simulations drops stronger with increasing wave number than observed. This suggests, that the spectral tails are damped in the simulations consistent with Skamarock (2004) and Larsén et al. (2012).

For wind shear, it is of interest how the simulated $\alpha$ differ from 0.11, which is used in the IEC maximal wind speed model. While the IQR of the $\alpha$ values varies little between the different simulations, the percentage of $\alpha$ values larger than 0.11 depends on the simulation, as well as the specific parts of a tropical cyclone. For the YSU simulation, $\alpha$ is mostly smaller than 0.11 for the analyzed scenes, namely before being affected significantly by land. Only 3.8 % (0.9 %) of the values are

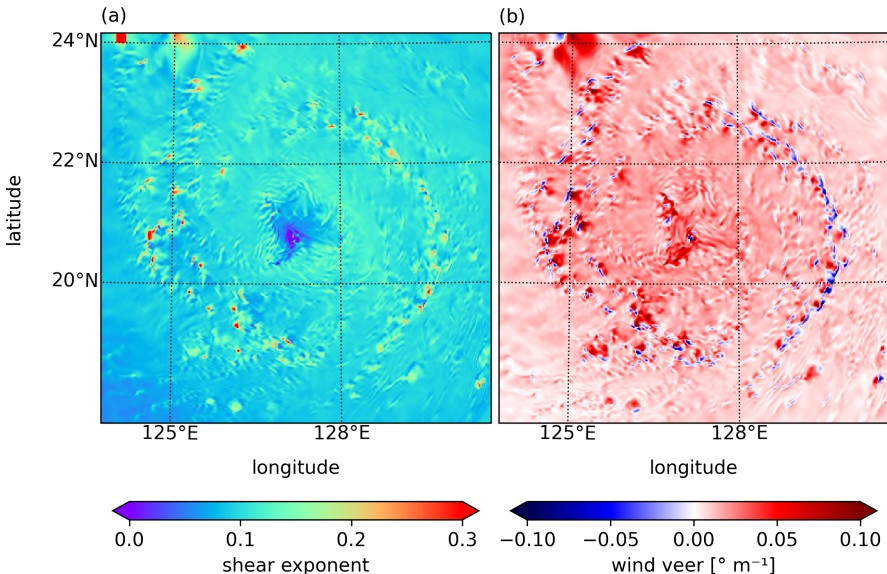

**Figure 12.** Horizontal fields taken from the MYJ model output at 00 UTC on 26 September of a) the shear exponent and b) wind veer.

larger than 0.11 in the outer cyclone (eyewall) in the YSU simulation. The fraction is larger for the MYNN simulation. MYNN produces 10.4 % of the $\alpha$ values larger than 0.11 in the outer cyclone region. For MYJ with the largest median $\alpha$, a large fraction of $\alpha$ exceeds 0.11: the percentages larger than 0.11 are 43.6 % in the eyewall and 22.3 % in the outer cyclone region.
The tails of the $\alpha$ distribution include values larger than 0.11. In the eyewall of the MYJ simulation, 14.2 % of the $\alpha$ values are larger than 0.12, and 0.7 % are larger than 0.15. In the outer cyclone region, 12.6 % of the $\alpha$ values are larger than 0.12, 4.1 % are larger than 0.15, and 1.0 % are larger than 0.2.

For all the simulations, wind veer is mostly confined within 0 and 0.03 $^\circ\,\mathrm{m}^{-1}$. However, the tails of the distribution are thick. To show where large $\alpha$ and veer values occur in the tropical cyclone, the horizontal field of $\alpha$ and veer at 00 UTC on
26 September is shown in Fig. 12. In all simulations, the spatial distribution of shear and veer is not uniform and varies between different regions of the tropical cyclone. In the outer cyclone region, the maximal values of shear and veer are found along the spiraling rainbands (Fig. 6). Along the rainbands, there is a zone of lower horizontal wind speed. Within this zone, local maxima and minima of wind shear are alternating. The wind veer changes from positive values (inflow angle decreasing with height) on the radially inward side of the rainbands to negative values (inflow angle increasing with height) on the outside.
To further assess the horizontal asymmetry, wind speed, shear, and veer are analyzed in four quadrants relative to the storm motion. Median values are listed for the MYJ simulation in each quadrant in Table 3. Note, that the MYJ simulation resulted in the most severe wind speed, shear, and veer. In the Eyewall wind speed, shear and veer show less variation between the quadrants than between the different simulations. However, the asymmetry of the wind field reflects in the area forming the eyewall region in each quadrant. The two right quadrants have a larger eyewall area than the two left quadrants. The difference
in the area is 65 % between the larger eyewall area in the back right quadrant and the smaller eyewall area in the front left

**Table 3.** Median of wind speed at $139\,\mathrm{m}$, wind shear exponent, and wind veer, as well as the percentage of shear exponent values larger than 0.11. The values are listed for the eyewall region and outer cyclone region for four motion-relative storm quadrants for the MYJ simulation. The area falling into the eyewall and outer cyclone region is further listed for each quadrant.

| | | | Wind speed | Shear exponent | | Wind veer |
| --- | --- | --- | --- | --- | --- | --- |
| Region | Quadrant | region size [$\mathrm{km}^2$] | median [$\mathrm{m\,s}^{-1}$] | median | % > 0.11 | median [$^\circ\,\mathrm{m}^{-1}$] |
| | front right | $6.2 \times 10^4$ | 39.4 | $1.1 \times 10^{-1}$ | 41.0 | $1.7 \times 10^{-2}$ |
| | front left | $3.9 \times 10^4$ | 38.7 | $1.0 \times 10^{-1}$ | 29.2 | $1.8 \times 10^{-2}$ |
| Eyewall | back left | $4.9 \times 10^4$ | 39.7 | $1.1 \times 10^{-1}$ | 41.0 | $1.6 \times 10^{-2}$ |
| | back right | $7.2 \times 10^4$ | 39.9 | $1.1 \times 10^{-1}$ | 53.4 | $1.6 \times 10^{-2}$ |
| | front right | $2.2 \times 10^5$ | 27.0 | $1.0 \times 10^{-1}$ | 31.1 | $1.3 \times 10^{-2}$ |
| | front left | $2.4 \times 10^5$ | 24.1 | $9.2 \times 10^{-2}$ | 14.6 | $1.2 \times 10^{-2}$ |
| Outer cyclone | back left | $2.2 \times 10^5$ | 24.6 | $9.5 \times 10^{-2}$ | 17.0 | $1.1 \times 10^{-2}$ |
| | back right | $1.9 \times 10^5$ | 28.4 | $1.0 \times 10^{-1}$ | 27.3 | $1.0 \times 10^{-2}$ |

quadrant. Different from the eyewall region, in the outer cyclone wind speed shear and veer show clear variations between the quadrants. Wind speed and wind shear are larger in the right quadrants than in the left quadrants. The difference in the median wind speed is $4.3\,\mathrm{m\,s}^{-1}$ between the front left and the back right quadrant. For $\alpha$ the difference is $8.6 \times 10^{-3}$ between the two right sectors and the back left sector. In the back right sector, 27.3 % of the profiles have $\alpha$ values larger than 0.11. Different from wind shear, wind veer is larger in the two front quadrants than in the back quadrants. The largest median wind veer, in the front right quadrant, is $1.3 \times 10^{-2}$.

## 4 Discussion

The three WRF simulations using the MYNN, YSU, and MYJ boundary layer schemes can produce a typhoon with a physically realistic track in terms of propagation speed and direction. Track position discrepancies are within $130\,\mathrm{km}$. As the track of the ERA5 reanalysis data shows good agreement with the best track data sets, it is evident that the track errors develop within the simulations. The higher SLP at the end of the MYNN simulation indicates, that this simulation underestimates Megi's intensity. However, this doesn't directly lead to the conclusion that the MYNN boundary layer scheme produces overly weak cyclones in general. On one hand, the best track data sets are mainly based on satellite observation, without direct in-situ measurements. This leads to uncertainty in the data as reflected by the spread of the JMA and JTWC data sets. On the other hand, the results of this study cannot readily be generalized to tropical cyclones with different intensities and storm sizes. At last, accounting for atmosphere, ocean, and wave interactions may further improve model performance and simulated tropical cyclone intensity.

Given that many factors play into the model results, one may question how representative the selected case is, regarding the effect of the boundary layer scheme. We argue that the differences between the MYNN simulation on the one hand and the YSU and MYJ simulation on the other hand strongly agree with documented sensitivity studies. Similar to our case, Rajeswari et al. (2020) found weaker storms for MYNN than for YSU, and most intense storms for MYJ, related to weaker low-level inflow in the MYNN scheme for five cyclones over the Bay of Bengal using WRF version 3.8. MYNN's lower intensity most likely relates to higher vertical diffusion in the MYNN scheme, which has been found to result in less intense storms, larger radius of maximal wind speed, and weaker radial inflow (Kepert, 2012; Gopalakrishnan et al., 2013; Zhang et al., 2020). The differences in the radius of maximal wind speed between the three simulations (see Fig. 7) most likely relate to the cyclone intensity and eddy diffusivity (Gopalakrishnan et al., 2013; Zhang et al., 2020).

In the extreme wind speed model (Eq. 1), the IEC standard proposes to use a $\alpha$ of $1.1 \times 10^{-1}$ (IEC, 2019a). The simulated $\alpha$ is in the median equal to or less than $1.1 \times 10^{-1}$ for all boundary layer schemes. With that, the median shear for extreme wind defined in the IEC standard is as steep or steeper when compared to the mesoscale simulation of typhoon Megi before landfall. This is in agreement with the $\alpha$ values obtained from Vickery et al. (2009). The simulated median $\alpha$ are similar in their order of magnitude and their sensitivity to the boundary layer scheme to simulations during neutral atmospheric stability at an offshore location in Denmark (Krogsæter and Reuder, 2014). In neutral atmospheric stability Krogsæter and Reuder (2014) find that the YSU scheme produces simulation with smaller $\alpha$ ($7.7 \times 10^{-2}$) compared to the MYNN ($8.8 \times 10^{-2}$) and the MYJ ($1.09 \times 10^{-2}$) scheme, similar to the results of our study. Further, the simulated wind veer is relatively small in comparison with wind veer found in low-wind regimes and particularly during stable conditions. This supports the conclusion, that the current IEC standard may be sufficiently similar in terms of wind shear and veer during tropical cyclones over open water. However, there are clear limitations to this conclusion as discussed in the following.

1. Larger values for $\alpha$ and veer are found in LES simulations over open water. Sanchez Gomez et al. (2023) find in LES simulations, that the mean $\alpha$ is about 0.2 near the eyewall. Both Kapoor et al. (2020) and Sanchez Gomez et al. (2023) find significant wind veer in LES simulations. Concretely, Sanchez Gomez et al. (2023) report a mean wind veer between $5.3 \times 10^{-2\,\circ}\,\mathrm{m}^{-1}$ and $6.9 \times 10^{-2\,\circ}\,\mathrm{m}^{-1}$. The differences between the study from Sanchez Gomez et al. (2023) and the simulated wind veer in our simulations can come from higher resolved wind veer variability in LES simulations or an overall shifted wind veer distribution due to differences in the mean wind field. In fact, Li et al. (2021) and Ren et al. (2022) find that the inflow layer was shallower and stronger in LES simulation compared to mesoscale simulations. Such a stronger, shallower inflow layer directly leads to a larger mean wind veer.

2. The analysis is based on mesoscale simulation and cannot resolve scales smaller than 15 km. With that structures such as large-scale vortices are not resolved. Such unresolved structures may contribute to enhanced shear and veer.

3. The study analyzes a typhoon case over the open ocean, before being affected significantly by land. In contrast to our simulations over the open ocean, He et al. (2016) and Tse et al. (2013) use wind observations in coastal areas. Both studies find wind shear larger than in the current IEC standard during typhoon conditions. He et al. (2016) find $\alpha$ in the range of 0.152 to 0.175 for profiles with marine exposures during 22 typhoons over Hong Kong. Tse et al. (2013)

find $\alpha$ values of 0.14 to 0.25 during typhoon Fengshen and Molave for profiles with marine exposure. The larger wind shear in these two studies could be a suggestion that wind shear may increase during the landfall of a tropical cyclone. Further studies are needed to understand how wind shear and veer evolve during landfall. Similarly, He et al. (2016) finds wind veer on the order of $2.8 \times 10^{-2\,\circ}\,\mathrm{m}^{-1}$ from the surface to the height of maximal wind speed from wind direction measurements with an open water fetch over a coastal area. This is around $0.003^\circ\,\mathrm{m}^{-1}$ larger than the wind shear in the YSU and MYJ simulation between the surface and $800\,\mathrm{m}$ (the height of simulated maximal wind speed) and around $0.01\,^\circ\,\mathrm{m}^{-1}$ larger than in the MYNN simulation. The difference between the studies may originate from the different locations with respect to land.

4. The fraction of profiles with $\alpha$ larger than 0.11 is substantial in the MYJ simulation. In the eyewall region of the MYJ simulation, 43.6 % of the $\alpha$ values are larger than 0.11. Values up to 0.15 are reached in 0.7 % of the eyewall and 4.1 % outer cyclone region. In the outer cyclone region, 1.0 % of the $\alpha$ values are larger than 0.2. Such large $\alpha$ values in the tails of the distribution impact the wind speed above and below $z_{hub}$ in the extreme wind speed model (Eq. 1). As an example, we take the extreme wind speed model with a $z_{hub}$ of 140 m, and a $V_{ref}$ of 57 $\mathrm{m\,s}^{-1}$. The wind speed from the extreme wind speed model at 180 m is 0.6 $\mathrm{m\,s}^{-1}$ (1.3 $\mathrm{m\,s}^{-1}$) larger with an $\alpha$ of 0.15 (0.20) than with an $\alpha$ of 0.11.

5. The analysis of the shear and veer distribution is sensitive to the definition of the outer cyclone and eyewall region. When restricting the definition of the eyewall region to a narrower band with a width of $0.4 \times$ RMW, the median wind speed in the eyewall increases on the order of 12 %, the shear exponent 4 %, and veer 10 %.

6. Wind veer and wind shear vary between the different sectors. Using the extreme wind speed model the difference in median $\alpha$ between the four quadrants leads to differences in the wind speed at above and below the hub height. For example, with a $z_{hub}$ of 140 m and a $V_{ref}$ of 57 $\mathrm{m\,s}^{-1}$, the difference at 180 m between the quadrants in the MYJ simulation is 0.2 $\mathrm{m\,s}^{-1}$.

7. Veer and $\alpha$ are larger close to the surface. In particular, wind veer is up to 18 % larger at the rotor bottom than at hub height in the MYJ simulation. In the extreme wind speed model $\alpha$ is assumed to be constant. Using two different $\alpha$ values below and above the hub height instead of assuming a constant $\alpha$, the median wind speed at the rotor bottom varies up to 0.5 $\mathrm{m\,s}^{-1}$ in the eyewall region. In the outer cyclone region, the median difference is up to 1.5 $\mathrm{m\,s}^{-1}$.

The simulated profile structure qualitatively agrees with the structure of dropsonde measurements from Vickery et al. (2009). However, the structure of the jet nose in the YSU simulation differs from their engineering model (Eq. 6). The engineering model describes a decreasing slope in a semi-log plot with height. Differently, the profiles of the YSU simulation have an increase in their slope below the height of the jet. This can be seen in Fig. 8 between $250\,\mathrm{m}$ to $800\,\mathrm{m}$ ($300\,\mathrm{m}$ to $1000\,\mathrm{m}$) in the eyewall (outer cyclone) region. However, it is not given that the empirical formulation holds for typhoon Megi because the MBL wind speed in the YSU simulation is larger than $40\,\mathrm{m\,s}^{-1}$. For such high wind speed profiles, an increase in slope with height below the jet nose is detectable in Fig. 2 and 8 in Vickery et al. (2009). Likewise, He et al. (2022) observed that the curvature of the typhoon wind profile is larger than predicted by the logarithmic law at heights of around 200 m. However,

they suggest that this relates to an internal boundary layer forming over land. This enhanced vertical wind speed gradient in the YSU simulation is located at heights not relevant for current wind turbines. More relevant for the analysis of shear is a small derivation from the logarithmic wind profile in the MYNN simulation. The slope of the semi-log wind profile is higher between the first and the second model layer than above. This change in slope was more pronounced in model runs with larger vertical grid spacing (not shown). For completeness, we report on finding discontinuous profiles with the MYJ boundary layer scheme in simulations with more vertical model levels (WRF version 3.7.1, fixed domains, 80 vertical layers). The discontinuities were found between 40 and 300 m and appeared to be related to the fine vertical grid spacing. This gave an incentive to lower the number of vertical levels in the current study.

The maximal 10 m wind speed of the simulations lies within the given values of the two best track data sets. This shows that the maximum of the 10 m wind speed from a $2\,\mathrm{km}$ grid is a valid approximation for the maximal ten-minute sustained wind field. In contrast, the maximal wind speed in the ERA 5 reanalysis has lower maximal wind speeds. Following Nolan et al. (2009b), a relative agreement between the simulated maximal wind speed and the ten-minute sustained wind field in the studied simulations is expected as explained subsequently. As seen in Fig. 11, the horizontal variability can be reproduced to scales of around $X = 15$ km. The fastest simulated wind speeds at $10\,\mathrm{m}$ are on the order of $WS = 40\ \mathrm{m\,s}^{-1}$ (see Fig. 7). The associated resolved simulated temporal resolution corresponds to $X/WS \simeq 6$ minutes. The effective spatial resolution of approximately 15 km corresponds to $7.5 \times$ the horizontal grid spacing (2 km in our simulations) in agreement with Skamarock (2004). The loss of variability on scales smaller than 15 km is related to horizontal diffusion in WRF (Skamarock, 2004).

The spectral slope of minus five-thirds found for both simulations and SAR data for wavelength larger than 15 km, agrees with Gage and Nastrom (1986). At smaller wavelengths, the spectral energy density in the SAR products increases with decreasing wavelength. This can be attributed to the superposition of three-dimensional turbulence on top of the mesoscale quasi-two-dimensional turbulence (Karagali et al., 2013; Larsén et al., 2016). The mean magnitude of the SAR spectra agrees best with the MYJ spectra for wavelengths larger than 15 km. However, the magnitude of the SAR spectra depends on what typhoon area is covered in the SAR image and on the analyzed time step. Because the area differs from the area covered by the simulation domain, comparing their magnitude provides only limited insight. Furthermore, the mean wind speed obtained by the SAR product might be subject to uncertainty, as SAR calibration over extreme wind areas is rare. In fact, SAR products with significantly higher wind speeds are given by Jackson et al. (2021). These were in good agreement with the JTWC best track data, as opposed to the here-used SAR product being closer to the JMA data set, which of course depends on the algorithms for the specific SAR retrievals for their case.

## 5    Conclusions

Due to the potentially large influence on wind turbine loads, we analyze hub height wind speed, wind shear, and wind veer over the rotor plane. The horizontal distribution and variability of these parameters are analyzed in the eyewall and outer cyclone region of typhoon Megi (2016) using a mesoscale modeling framework. To evaluate model uncertainty related to the boundary

layer parametrization, three frequently used boundary layer schemes (MYJ, MYNN, and YSU) are analyzed in WRF(version 4.4).

Our analysis showed that:

1. All three simulations can reasonably reproduce the typhoon track and cyclone structure. The storm intensifies in all model realizations and the spread of the simulated storm intensity is comparable to the spread between best track data sets. With that, the spread between the models in wind speed, wind shear, wind veer and their variability can be regarded as model uncertainty.

2. The simulated hub height wind speed is sensitive to the boundary layer parametrization and its median varies between the schemes by 15 % (8 %) in the eyewall (outer cyclone) region.

3. Regardless of the boundary layer parametrization the simulated median wind shear exponents ($\alpha$) is smaller or equal to 0.11 used in the extreme wind model in the IEC standards (IEC, 2019a). However, in the MYJ simulation, 43.6 % of the wind profiles in the eyewall region exceed 0.11. The simulated median wind shear is in good agreement with the study from Vickery et al. (2009), but smaller than observed in coastal areas during tropical cyclone conditions (Tse et al., 2013; He et al., 2016). The difference in the median $\alpha$ between the simulations using different boundary layer schemes is $2.5 \times 10^{-2}$ ($2.9 \times 10^{-2}$) in the eyewall region (outer cyclone region). This difference is small compared to the difference between the simulated offshore typhoons and observations over coastal areas (Tse et al., 2013; He et al., 2016).

4. Median wind veer is up to $1.7 \times 10^{-2\,\circ}\,\mathrm{m}^{-1}$ ($1.2 \times 10^{-2\,\circ}\,\mathrm{m}^{-1}$) in the eyewall (outer cyclone). It is up to 18 % larger at the rotor bottom than at hub height. The simulated wind veer is relatively small compared to wind veer in moderate wind speed regimes. This stands in contrast to studies from Worsnop et al. (2017), Kapoor et al. (2020) and He et al. (2016) who found strong wind veer in tropical cyclone LES. The difference in wind veer between the simulations with different boundary layer schemes is $3.0 \times 10^{-3\,\circ}\,\mathrm{m}^{-1}$ ($2.5 \times 10^{-3\,\circ}\,\mathrm{m}^{-1}$) in the eyewall (outer cyclone) region.

5. Spectral analysis of the three simulations with a $2\,\mathrm{km}$ horizontal grid spacing shows, that horizontal wind speed variability is resolved on scales larger than $15\,\mathrm{km}$ as expected. The variability is largest in the MYJ simulation followed by the YSU simulation and smallest in the MYNN simulation. The produced horizontal difference in variability between the three schemes is evident over the analyzed wavelength range (4-100 km). Because the spectral energy density decays less with increasing wavenumber in the MYJ scheme, the difference in horizontal wind speed variability is most pronounced at the smallest resolved wavelengths.

6. Overall hub hight wind speed, wind shear, and wind veer are larger in the eyewall region than in the outer cyclone region. Within the outer cyclone region, a clear spatial organization of wind shear and veer is found along the spiraling rainbands. In fact, local maxima in shear and veer along the rainbands are larger than the maximal simulated values in the eyewall region. On the radially inward side of the rainbands, positive veering angles reach their maximum over the tropical cyclone structure. On the outward side, either small wind veering angles or backing (negative wind veer) dominate.

Wind shear maxima and minima on scales of around 20 km alternate along the rainband. This spatial organization likely

leads to rapid coherent changes in the wind profile at a possible wind turbine location.

Based on these conclusions, further investigation is needed to address 1) how wind speed, shear, and veer in tropical cyclones evolve during landfall, 2) how much wind shear and wind veer vary between tropical cyclones with different intensities and radii, and 3) how much wind turbine load estimates based on the current IEC standard differ between load estimates based on the simulated wind speed, shear, and veer distributions.

*Code and data availability.* The source code of the WRF model is available at https://www.mmm.ucar.edu/models/wrf (Skamarock et al., 2021). WRF namelists used for the simulations are available at https://zenodo.org/records/10202995 (Müller, 2023). The ERA 5 data used as boundary conditions for the WRF simulation is available at https://doi.org/10.24381/cds.bd0915c6 (Hersbach et al., 2018). The JTWC best track data set is available at https://www.metoc.navy.mil/jtwc/jtwc.html?best-tracks (Chu et al., 2002) and the JMA best track data set is available from https://www.jma.go.jp/jma/jma-eng/jma-center/rsmc-hp-pub-eg/trackarchives.html. The SAR data is available at

https://doi.org/10.11583/DTU.19704883.v1 (Badger et al., 2022).

*Author contributions.* SM, XGL planned the study; SM performed the model simulations with input from XGL; SM conducted the model analysis with input from XGL and DV. SM wrote the manuscript draft; XGL, DV reviewed and edited the manuscript.

*Competing interests.* The authors declare that no competing interests are present.

*Acknowledgements.* This study is supported by the SDC project 906421. The authors acknowledge the computational and data resources
provided on the Sophia HPC Cluster at the Technical University of Denmark (DTU), DOI: 10.57940/FAFC-6M81. We thank Jana Fischereit and Ebba Dellwik from DTU for the feedback on the paper draft. We further thank Marc Imberger and Oscar Manuel Garcia Santiago from DTU for their help and discussions.

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
