# Peer review of "Tropical cyclone low-level wind speed, shear, and veer: sensitivity to the boundary layer parameterization in WRF"

_Wind Energy Science, 2023_

## Referee Comment (RC1)

**Review: Tropical cyclone low-level wind speed, shear, and veer: sensitivity to the boundary layer parameterization in WRF**

**Summary**

The authors perform a study evaluating how different boundary layer parameterizations in WRF modify mean wind characteristics relevant for offshore wind energy. The authors simulate typhoon Megi using the YSU, MYJ, and MYNN boundary layer schemes. The simulation results are validated using best-track estimates, surface wind speed observations, and synthetic aperture radar data. After validation, wind speed, wind speed shear, and wind veer are estimated over the turbine rotor layer of a hypothetical 14MW wind turbine in the eyewall vicinity and in the rainband region. The authors contrast current wind turbine design standards against mean wind characteristics in the boundary layer for each boundary layer scheme. Wind speed in the rotor layer varies with boundary layer parameterization, especially near the eyewall. Wind shear and veer also vary, but to a lesser extent. In general, the authors report low wind veer in the turbine rotor layer and shear that agrees with turbine design standards.

The manuscript provides a coherent story and is an important addition to the growing literature on tropical cyclones. However, there are major concerns with the manuscript that should be addressed before publication.

**Major comments:**

1. Velocity fields are averaged azimuthally: The velocity fields in cyclones have distinct characteristics depending on the azimuthal location (Ren et al., 2019). For instance, the height of the tangential wind speed maxima and the height of the inflow varies azimuthally. Therefore, it is possible that by performing this azimuthal averaging, these characteristics are being lost, and shear and veer are being underestimated.
2. Radial locations in analysis: The radial distance that defines the eyewall regions is very large (60 km), thus the most extreme wind conditions that occur in the vicinity of the radius of maximum winds might be eclipsed by slower winds inside the eyewall. How is the 60 km to 120 km range chosen to define this as the eyewall region? Furthermore, given that the different PBL schemes result in different storm sizes, the radial distances might not be entirely equivalent for the different storms.
3. Context: The authors do not acknowledge the role of large-scale turbulence structures that form in tropical cyclones and the limitation of their simulations in resolving these structures. Please clarify that your simulations do not accurately resolve structures smaller than 15 km in wavelength due to the employed grid resolution. As a result, small- and large-scale variability in the boundary layer relevant for wind turbine design (such as large-scale vortices) are not captured.
4. A brief description of wind turbine design standards should be provided. The authors should include a brief explanation of the extreme wind speed models used for design load cases 6.1-6.4 in the IEC 61400-3 standard.

**Minor comments:**

Please review English writing throughout the manuscript. Some examples include:
- Line 48: Hyphen in large-eddy simulations.
- Line 70: Capitalize Weather Research and Forecasting model.

Line 22: Please include the role of large-eddy simulations, which can resolve large- and small-scale turbulence structures (e.g., mesovortices) relevant for loads in wind turbines.

Line 48: depending on the grid resolution (e.g., 2 km), mesoscale simulations might not resolve scales smaller than 10 km either.

Line 117: Is the timestep the same for all domains? A 45 second timestep for the 2 km domain seems too long, especially for such intense storm.

Section 2: Please mention vertical grid resolution in the lowest levels.

Line 74: Please clarify the spinup time for each domain. Is it 12 h for domain 1, or are all domains initialized at the same time?

Line 125: Please clarify that veer is the shortest rotational path of the wind vector and, as such, is restricted to $|\text{Veer}| \leq 180°$.

Line 139: Note that the averaging period is very different between Kapoor et al. (2020) and your simulations. Therefore, wind veer is expected to differ.

Section 2.2: Is there a reason for restricting the shear analysis to two heights only? Did the authors consider fitting wind speed at all model heights within the rotor layer to the power-law wind profile to estimate $\alpha$?

Section 3.2: Please report the mean radius of maximum winds for each boundary layer scheme.

Line 206: Minimum SLP and maximum wind speed are shown in Figure 4.

Caption Figure 4: The 0.93 factor converts from 1-min to 10-min averaged winds and not the other way around.

Figure 5: Consider moving panels d,e,f to a new figure farther down in the manuscript. The authors only comment on these panels in line 255, after referring to Figure 6, 7, and 8.

Figures 1, 2, 3, 4, 5, 8: Please include axes and color bar labels where needed.

Lines 221-225: Please comment on the limitations of comparing instantaneous velocities from the mesoscale domains with 1-min and 10-min observational averages. These are not entirely comparable and the instantaneous velocity fields are grid dependent.

Line 231: Please clarify what you mean with symmetric wind component. Is this the tangential wind speed?

Line 235: Units for 12 m s$^{-1}$.

Line 253: Consider relocating numbers in sentence for clarity.

Line 257: Please rephrase for clarity. Perhaps break down into two sentences.

Line 269: Please clarify that the simulations only resolve large-scale variability of atmospheric variables.

Section 4: Please comment on the limitations of estimating veer from these simulations given that the depth and intensity of the radial inflow varies with grid resolution (Xu et al., 2021; Ren et al., 2022).

Line 312: Please soften the language in this sentence. Shear in the tropical cyclone boundary layer has been shown to be different in LES and mesoscale simulations (Ren et al., 2022; Xu et al., 2021; Li et al., 2021).

Line 312: This conclusion is drawn based on median wind characteristics. What about wind characteristics that are near the tail of the distribution (e.g., 0.75 percentile)?

Figure 7, Figure 8, and Table 2: The wind profiles representing the 0.75 percentile in Figure 7 display much larger shear than the median wind profiles. This is also evident in the inter quartile range for $\alpha$ reported in Table 2 and in the distribution of $\alpha$ in Figure 8. Please comment on the percentage of wind profiles with shear exponent larger than 0.11. Based on Table 2, it seems about 25% of wind profiles may display shear larger than 0.11.

Figure 8: Why are the y-axis tick labels for shear and veer larger than 1 if this is a plot of probability density?

**References**

Kapoor, A., Ouakka, S., Arwade, S. R., Lundquist, J. K., Lackner, M. A., Myers, A. T., Worsnop, R. P., and Bryan, G. H.: Hurricane eyewall winds and structural response of wind turbines, Wind Energ. Sci., 5, 89–104, https://doi.org/10.5194/wes-5-89-2020, 2020.

Li, X., Pu, Z., and Gao, Z.: Effects of Roll Vortices on the Evolution of Hurricane Harvey During Landfall, Journal of the Atmospheric Sciences, https://doi.org/10.1175/JAS-D-20-0270.1, 2021.

Ren, H., Dudhia, J., Ke, S., and Li, H.: The basic wind characteristics of idealized hurricanes of different intensity levels, Journal of Wind Engineering and Industrial Aerodynamics, 225, 104980, https://doi.org/10.1016/j.jweia.2022.104980, 2022.

Ren, Y., Zhang, J. A., Guimond, S. R., and Wang, X.: Hurricane Boundary Layer Height Relative to Storm Motion from GPS Dropsonde Composites, Atmosphere, 10, 339, https://doi.org/10.3390/atmos10060339, 2019.

Xu, H., Wang, H., and Duan, Y.: An Investigation of the Impact of Different Turbulence Schemes on the Tropical Cyclone Boundary Layer at Turbulent Gray-Zone Resolution, JGR Atmospheres, 126, https://doi.org/10.1029/2021JD035327, 2021.

---

## Author Comment (AC1)

**Answer to reviewer comments**

November 24, 2023

**1 Summary of the updates**

Dear reviewers, we would like to thank you for the constructive and concrete feedback on our manuscript. We highly appreciate your input. Based on your comments we revised our manuscript. Major updates are made, including:

1. We revised the definition of the eyewall and rainband section.

2. We include profiles of shear and veer.

3. We include a short description of the IEC extreme wind speed model.

4. We update the introduction part on subgrid scale parametrization.

5. We update the description of the boundary layer schemes.

6. We elaborate on differences between different storm sectors.

7. We highlight limitations on the generally low wind shear and veer encountered in our simulations.

In the following, we respond point-by-point to your comments and refer to the updates in the manuscript.

**2 Reviewer comments from the first referee**

**2.1 Summary**

The authors perform a study evaluating how different boundary layer parameterizations in WRF modify mean wind characteristics relevant for offshore wind energy. The authors simulate typhoon Megi using the YSU, MYJ, and MYNN boundary layer schemes. The simulation results are validated using best-track estimates, surface wind speed observations, and synthetic aperture radar data. After validation, wind speed, wind speed shear, and wind veer are estimated over the turbine rotor layer of a hypothetical 14MW wind turbine in the eyewall against mean wind characteristics in the boundary layer for each boundary layer scheme. Wind speed in the rotor layer varies with boundary layer parameterization, especially near the eyewall. Wind shear and veer also vary, but to a lesser extent. In general, the authors report low wind veer in the turbine rotor layer and shear that agrees with turbine design standards.

Thank you for this summary.

**2.2 Mayjor comments:**

1. Velocity fields are averaged azimuthally: The velocity fields in cyclones have distinct characteristics depending on the azimuthal location (Ren et al., 2019). For instance, the height of the tangential wind speed maxima and the height of the inflow varies azimuthally. Therefore, it is possible that by performing this azimuthal averaging, these characteristics are being lost, and shear and veer are being underestimated.

   Thank you for raising this point. Based on your comment, we looked into the asymmetries over the tropical cyclone azimuth in detail. We address the variability between different sectors now in the results and discussion sections. We further changed the definition of the eyewall region, as described in the answer to your second comment. The new eyewall definition is no longer based on axisymmetry. With this new definition, the difference in the wind speed, shear, and veer between the storm quadrants is reduced in the eyewall region.

   We now report on differences between the quadrants as follows (Please refer to the manuscript for Table 3):

**In the result section:** *To further assess the horizontal asymmetry, wind speed, shear, and veer are analyzed in four quadrants relative to the storm motion. Median values are listed for the MYJ simulation in each quadrant in Table 3. Note, that the MYJ simulation resulted in the most severe wind speed, shear, and veer. In the Eyewall wind speed, shear and veer show less variation between the quadrants than between the different simulations. However, the asymmetry of the wind field reflects in the area forming the eyewall region in each quadrant. The two right quadrants have a larger eyewall area than the two left quadrants. The difference in the area is 65 % between the larger eyewall area in the back right quadrant and the smaller eyewall area in the front left quadrant. Different from the eyewall region, in the outer cyclone wind speed shear and veer show clear variations between the quadrants. Wind speed and wind shear are larger in the right quadrants than in the left quadrants. The difference in the median wind speed is $4.3\,\mathrm{m\,s^{-1}}$ between the front left and the back right quadrant. For $\alpha$ the difference is $8.6 \times 10^{-3}$ between the two right sectors and the back left sector. In the back right sector, $27.3\,\%$ of the profiles have $\alpha$ values larger than 0.11. Different from wind shear, wind veer is larger in the two front quadrants than in the back quadrants. The largest median wind veer, in the front right quadrant, is $1.3 \times 10^{-2}$.*
**In discussion:** *The fraction of profiles with $\alpha$ larger than 0.11 is substantial in the MYJ simulation. In the MYJ simulation 43.6 % (22.3 %) of the values in the eyewall region (outer cyclone region) are larger than 0.11. This portion is even larger for the right back quadrant of the MYJ simulation, where the median is larger than 0.11 and 53.3 % are larger than 0.11.*

2. Radial locations in analysis: The radial distance that defines the eyewall regions is very large (60 km), thus the most extreme wind conditions that occur in the vicinity of the radius of maximum winds might be eclipsed by slower winds inside the eyewall. How is the 60 km to 120 km range chosen to define this as the eyewall region? Furthermore, given that the different PBL schemes result in different storm sizes, the radial distances might not be entirely equivalent for the different storms.

This is a valid point. We decided to redefine the eyewall region and base the definition now on wind speed. The new criteria are described as follows in the method section (please refer to Fig. 2 in the manuscript):

*The eyewall and outer cyclone regions are analyzed separately. With that, we can account for and characterize differences between the two storm regions. To avoid, that the position of the simulation domain relative to the cyclone center influences the analysis, we use only grid points within a distance to the cyclone center (R) smaller than 350 km. The definitions of the eyewall and outer region are illustrated in Fig. 2 and summarized in Eq. 4 and 5. We define the eyewall as a high wind speed regime. Our definition is based on the simulated wind speed at 10 m ($WS_{10}$) at each simulation time step. Grid points are assigned to the eyewall region if two criteria are fulfilled: 1.) $WS_{10}$ is greater than or equal to the 80th percentile of $WS_{10}$ ($P_{80}(WS_{10})$), and 2.) R is less than 250 km. The selected eyewall area is 76000 $\mathrm{km}^2$. The thickness of the eyewall is not symmetrical over the azimuth and can be zero. Note also, that the area includes gaps, where the wind speed is lower than $P_{80}(WS_{10})$. The outer region includes all grid points that are not part of the eyewall and the eye. To distinguish between the outer region and the eye, we define a critical radius $R_{Eye}$ in Eq. 5. Here, $R_{Eye}$ is taken as the 10th percentile of R of the eyewall grid points ($P_{10}(R_{Eyewall})$). Thus*

$$Region = \begin{cases} Eyewall: & if\ WS_{10} \geq P_{80}(WS_{10})\ and\ R < 250\,km, \\ Outer\ cyclone: & if\ not\ Eyewall\ and\ R > R_{Eye} \end{cases} \tag{1}$$

$$R_{Eye} = P_{10}(R_{Eyewall}) \tag{2}$$

*The outer region is not homogeneous. It includes both the rainbands and the non-convective moat areas. These areas have different properties. However, we do not separate them, to prevent the analysis from becoming sensitive to the selection criteria.*

3. Context: The authors do not acknowledge the role of large-scale turbulence structures that form in tropical cyclones and the limitation of their simulations in resolving these structures. Please clarify that your simulations do not accurately resolve structures smaller than 15 km in wavelength due to the employed grid resolution. As a result, small- and large-scale variability in the boundary layer relevant for wind turbine design (such as large-scale vortices) are not captured.

Good point - our arguments on this are somewhat hidden in the original version. We now highlight them at several locations in the manuscript.

**In the introduction:** *Typically they can only resolve wind speed variability on scales in the order of seven times the horizontal grid spacing (Skamarock, 2004). Smaller scale structures resolved in LES simulations, such as roll vortices, cannot be adequately resolved in mesoscale simulations (Li et al., 2021).*
**In the discussion:** *The analysis is based on mesoscale simulation and cannot resolve scales smaller than 15 km.*

*With that structures such as large-scale vortices are not resolved. Such unresolved structures may contribute to enhanced shear and veer.*

4. A brief description of wind turbine design standards should be provided. The authors should include a brief explanation of the extreme wind speed models used for design load cases 6.1-6.4 in the IEC 61400-3 standard.
Good point. A brief description of the IEC standard is added to the introduction. **New:** *To ensure the structural integrity of wind turbines, turbine design standards are defined in the International Electrotechnical Commission's standard (IEC) for onshore and offshore turbines (IEC, 2019a,b). The standards are based on site-specific wind speed classes and turbulence classes. The ability of wind turbines to withstand wind conditions within the turbine class is tested in design load cases (DLC). Different DLCs assess the loads acting on wind turbines during power production, and at stand-still. In and close to the eyewall wind speeds typically exceed the turbine-specific cutoff wind speed. In this case, turbines are parked to minimize loads. However, further away from the cyclone center, turbines may still be operating. Either way, the wind conditions tested in the DLCs consist of a mean wind profile combined with either a deterministic gust profile or turbulence. A power-law model is used for the wind profile with an associated hub height wind speed and wind shear over the rotor plane. A constant wind shear is suggested for load simulations of operating and parked turbines. Such a simplified wind shear model has an influence on turbine loading (Dimitrov et al., 2015). Wind veer, the change in wind direction with height is not accounted for in the IEC standards.*
**Old:** *The wind input for aeroelastic wind turbine models consists of a wind profile with an associated hub height wind speed and wind shear over the rotor plane. In the International Electrotechnical Commission's standard (IEC), a universal constant wind shear is suggested for load simulations (IEC, 2019a). Such a simplified wind shear model has an influence on turbine loading (Dimitrov et al., 2015). Wind veer, the change in wind direction with height is not accounted for in the IEC standards.*
In the Methods section, the extreme wind speed models are described. **New:** *The analysis of wind shear is performed, such that a comparison to the IEC standard is straightforward. For the assessment of extreme wind conditions, the IEC assumes a wind profile according to the extreme wind speed model:*

$$V_{50} = V_{ref} * (\frac{z}{z_{hub}})^\alpha \tag{3}$$

*Here $V_{50}$ is the extreme wind speed with a return period of 50 years, averaged over a ten minute interval. For areas affected by tropical cyclones, the reference wind speed $V_{ref}$ is $57 \, \mathrm{m\,s^{-1}}$. The height is given by $z$, and the hub height by $z_{hub}$. The wind shear exponent $\alpha$ is a measure of vertical wind shear. Strong wind shear is associated with larger $\alpha$ values. For extreme wind conditions, the IEC uses a constant $\alpha$ of 0.11 (IEC, 2019a). Under normal wind conditions offshore, $\alpha$ is set to 0.14 (IEC, 2019b).*
**Old:** *In the IEC, wind shear is mostly expressed as shear exponent $\alpha$, which describes a power-law wind profile (IEC, 2019a):*

**2.3   Minor comments:**

1. Please review English writing throughout the manuscript. Some examples include: - Line 48: Hyphen in large-eddy simulations. - Line 70: Capitalize Weather Research and Forecasting model.
Thank you, the corrections are made.

2. Line 22: Please include the role of large-eddy simulations, which can resolve large- and small-scale turbulence structures (e.g., mesovortices) relevant for loads in wind turbines.
The sentence is updated as follows: *Further research incorporating mesoscale **and micro-scale** numerical models into aeroelastic wind turbine models is necessary to achieve reliable structural analysis of wind turbines in tropical cyclone conditions (Li et al., 2022).*

3. Line 48: depending on the grid resolution (e.g., 2 km),mesoscale simulations might not resolve scales smaller than 10 km either.
We reformulate as follows: **New:** *Typically they can only resolve wind speed variability on scales in the order of seven times the horizontal grid spacing (Skamarock, 2004). Smaller scale structures resolved in LES simulations, such as roll vortices, cannot be adequately resolved in mesoscale simulations (Li et al., 2021).* **Old:** *Mesoscale atmospheric simulations use larger grid spacing than large eddy simulations and cannot resolve sub-kilometerscale variability (Skamarock, 2004).*

4. Line 117: Is the timestep the same for all domains? A 45 second timestep for the 2 km domain seems too long, especially for such intense storm. I am happy you have spotted this. The sentence is updated: *The three domains are run with a 45, 15, and 5 s timestep.*

5. Section 2: Please mention vertical grid resolution in the lowest levels.
The domain heights are added as follows: *All domains use 70 vertical layers. The lowest model levels have a mean height of 8, 26, 47, 72, 102, 139, 183, 234, 297, and 372 m.*

6. Line 74: Please clarify the spinup time for each domain. Is it 12 h for domain 1, or are all domains initialized at the same time?
The spin-up time is clarified as follows: *All three domains are initialized at 00 UTC on 25 September. The first 12 h are used as spin-up time. The following 36-hour simulations starting at 12 UTC on 25 September are used for the analysis.*

7. Line 125: Please clarify that veer is the shortest rotational path of the wind vector and, as such, is restricted to $|Veer| \leq 180°$.
The definition of wind veer is updated. We now decided to use a linear fit of the wind directions over all heights within the rotorplane instead of using only two heights. We chose to do this for two reasons: 1.) We would like to keep the definition of wind shear and wind veer similar (see point 9). 2.) We want to reduce the sensitivity of wind veer to the chosen model levels. We are aware that wind veer changes non-linearly with height. This limitation is discussed in the results and discussion sections. **New:** *Wind veer is defined as the shortest rotational path between the wind direction (WD) at different heights, which is maximal 180°. It is defined such that positive (negative) veer values describe clockwise (counterclockwise) rotation with increasing height. With that, a decreasing (increasing) inflow angle with height is associated with positive (negative) veer values. In this study we normalize wind veer by the vertical distance between two used heights. Veer is calculated as follows:*

$$Veer = \frac{\min(WD_2 - WD_1)}{z_2 - z_1} \tag{4}$$

*Here, WD is defined at the model heights $z_1$ and $z_2$. Similar to the analysis of $\alpha$, wind veer is analyzed between consecutive model levels at different heights, as well as over the rotor plane. For the latter, we use a least square fit between WD and z. All model levels between the rotor bottom and the rotor top are used for the fit.*

8. Line 139: Note that the averaging period is very different between Kapoor et al. (2020) and your simulations. Therefore, wind veer is expected to differ.
We decided not to comment on the assumption of a monotonic increase of shear and veer with height, and don't show the text passage. We decided this for two reasons: 1.) we now calculate wind shear using a linear fit, and 2.) we analyze the profile of $\alpha$ in the results section.

9. Section 2.2: Is there a reason for restricting the shear analysis to two heights only? Did the authors consider fitting wind speed at all model heights within the rotor layer to the power-law wind profile to estimate $\alpha$?
The difference between using two heights and a linear fit were small. We decided anyways to follow your comment and use a linear fit for the calculation. We further analyze the change of $\alpha$ with height in the result section. The definition is updated as follows: *In this study $\alpha$ is calculated using Eq. 2.*

$$\alpha = \frac{\ln(u_2/u_1)}{\ln(z_2/z_1)} \tag{5}$$

*Here, $u_1$ and $u_2$ are the wind speeds at heights $z_1$ and $z_2$ respectively. While $\alpha$ depends on height, the IEC assumes a constant $\alpha$ over the rotor plane. We analyze both $\alpha$ between consecutive model levels at different heights and the total $\alpha$ over the rotor plane. For the latter, we use a least square fit between $\ln(u)$ and $\ln(z)$. All model levels between the rotor bottom and the rotor top are used for the fit.*

10. Section 3.2: Please report the mean radius of maximum winds for each boundary layer scheme.
The radius is added: *Additionally, the distance between the eye and the maximal wind speed in the eyewall is larger in the MYNN (104 km) simulation than in the YSU (92 km) and the MYJ (94 km) simulations.*

11. Line 206: Minimum SLP and maximum wind speed are shown in Figure 4.
The reference to Figure 5 is added (the number of the figures increased by 1): *Megi's track and intensity in terms of minimal SLP and maximal wind speed from 25 September onward are shown in Fig. 4 **and 5.***

12. Caption Figure 4: The 0.93 factor converts from 1-min to 10-min averaged winds and not the other way around. The error is corrected: *Black crosses show the JTWC values multiplied by 0.93, as recommended by Harper et al. (2010) to convert one-minute sustained wind speeds to ten-minute sustained wind speeds in tropical cyclones over the sea.*

13. Figure 5: Consider moving panels d,e,f to a new figure farther down in the manuscript. The authors only comment on these panels in line 255, after referring to Figure 6, 7, and 8.
Thank you for the comment, the panels are moved.

14. Figures 1, 2, 3, 4, 5, 8: Please include axes and color bar labels where needed.
We updated all Figures to include Labels, unless axes are shared between subplots.

15. Lines 221-225: Please comment on the limitations of comparing instantaneous velocities from the mesoscale domains with 1-min and 10-min observational averages. These are not entirely comparable and the instantaneous velocity fields are grid dependent.
We commented as follows: *The comparison between the maximal wind speed and minimal SLP in the simulations and the best track data has shortcomings, especially since the two sizes depend on the spatial and temporal resolution of the simulation. Nevertheless, the comparison is widely used and helpful to qualitatively evaluate the simulated intensity (Rajeswari et al., 2020; Shenoy et al., 2021; Zhu et al., 2014).*

16. Line 231: Please clarify what you mean with symmetric wind component. Is this the tangential wind speed?
We agree that the word symmetric is not precise enough. We rewrite as follows: **New:** *Larger wind speeds are evident in the back right quadrant, northeastward of the eye. Here, wind speeds are around $4\,\mathrm{m\,s^{-1}}$ larger than in the back left quadrant, southwestward of the eye. Nevertheless, the wind speed changes similarly with increasing distance to the cyclone center in all quadrants.*
**Old:** *Larger wind speeds are evident in the top right quadrant, northwestward of the eye. Nevertheless, the symmetric wind speed component dominates over asymmetric features.*

17. Line 235: Units for 12 m s-1.
Thank you, we corrected it.

18. Line 253: Consider relocating numbers in sentence for clarity.
We reformulate as follows. **New:** *The larger slope in the MYJ wind speed profile with respect to YSU reflects in a $2.7 \times 10^{-2}$ to $2.8 \times 10^{-2}$ larger median shear exponent.*
**Old:** *The larger slope in the MYJ simulation with respect to the YSU reflects in $2.4 \times 10^{-2}$ ($3.0 \times 10^{-2}$) larger shear exponent in the exponent in the eyewall (rainband) region.*

19. Line 257: Please rephrase for clarity. Perhaps break down into two sentences.
The sentence is broken down into smaller sentences as follows. **New:** *The MYJ simulation has a $1.9\,\mathrm{m\,s^{-1}}$ larger eyewall wind speed at hub height than the YSU simulation (see Table 3). Note, that this is different at 10 m, where the two simulations have similar wind speeds in the eyewall. The difference between the two heights is a direct result of the larger shear exponent in the MYJ simulation with respect to the YSU simulation.*
**Old:** *With the larger gradient in wind speed with height, the MYJ simulation has a $1.9\,\mathrm{m\,s^{-1}}$ larger eyewall wind speed at hub height than the YSU scheme, while having a similar wind speed at 10 m (see Fig. 8 and Table 2).*

20. Line 269: Please clarify that the simulations only resolve large-scale variability of atmospheric variables.
The sentence is adjusted as follows: *Apart from the **Typhoon-scale** structure, the simulations also produce **mesoscale** variability in the wind field. **Mesoscale** wind fluctuations can be seen in all simulations within and outside of the eyewall in Fig. 9.*

21. Section 4: Please comment on the limitations of estimating veer from these simulations given that the depth and intensity of the radial inflow varies with grid resolution (Xu et al., 2021; Ren et al., 2022).
Thank you for the comment. We saw that Li et al. (2021) address the inflow layer depth and its sensitivity on grid resolution more directly than Xu et al. (2021). We add the following: *In fact, Li et al. (2021) and Ren et al. (2022) find that the inflow layer was shallower and stronger in LES simulation compared to mesoscale simulations. Such a stronger, shallower inflow layer directly leads to a larger mean wind veer.*

22. Line 312: Please soften the language in this sentence. Shear in the tropical cyclone boundary layer has been shown to be different in LES and mesoscale simulations (Ren et al., 2022; Xu et al., 2021; Li et al., 2021).

We admit that this sentence was ambitious. We soften the language slightly and list the limitations directly below.

**New:** *This supports the conclusion, that the current IEC standard **may be** sufficiently similar in terms of wind shear and veer during tropical cyclones. **However, there are clear limitations to this conclusion as discussed in the following.***

(a) ***Larger values for $\alpha$ and veer are found in LES simulations. Gomez et al. (2023) find in LES simulations, that the mean $\alpha$ is about 0.2 near the eyewall. Both Kapoor et al. (2020) and Gomez et al. (2023) find significant wind veer in LES simulations. Concretely, Gomez et al. (2023) report a mean wind veer between $5.3 \times 10^{-2\,\circ}\,\mathrm{m}^{-1}$ and $6.9 \times 10^{-2\,\circ}\,\mathrm{m}^{-1}$. The differences between the study from Gomez et al. (2023) and the simulated wind veer in our simulations can come from higher resolved wind veer variability in LES simulations or an overall shifted wind veer distribution due to differences in the mean wind field. In fact, Li et al. (2021) and Ren et al. (2022) find that the inflow layer was shallower and stronger in LES simulation compared to mesoscale simulations. Such a stronger, shallower inflow layer directly leads to a larger mean wind veer.***

(b) *The analysis is based on mesoscale simulation and cannot resolve scales smaller than 15 km. With that structures such as large-scale vortices are not resolved. Such unresolved structures may contribute to enhanced shear and veer.*

(c) *The study analyzes a typhoon case over the open ocean, before being affected significantly by land. In contrast to our simulations over the open ocean, He et al. (2016) and Tse et al. (2013) use wind observations in coastal areas. Both studies find wind shear larger than in the current IEC standard during typhoon conditions. He et al. (2016) find $\alpha$ in the range of 0.152 to 0.175 for profiles with marine exposures during 22 typhoons over Hong Kong. Tse et al. (2013) find $\alpha$ values of 0.14 to 0.25 during typhoon Fengshen and Molave for profiles with marine exposure. The larger wind shear in these two studies could be a suggestion that wind shear may increase during the landfall of a tropical cyclone. Further studies are needed to understand how wind shear and veer evolve during landfall. Similarly, He et al. (2016) finds wind veer in the order of $2.8 \times 10^{-2\,\circ}\,\mathrm{m}^{-1}$ from the surface to the height of maximal wind speed from wind direction measurements with an open water fetch over a coastal area. This is around $0.003^\circ\,\mathrm{m}^{-1}$ larger than the wind shear in the YSU and MYJ simulation between the surface and $800\,\mathrm{m}$ (the height of simulated maximal wind speed) and around $0.01^\circ\,\mathrm{m}^{-1}$ larger than in the MYNN simulation. The difference between the studies may originate from the different locations with respect to land.*

(d) *The fraction of profiles with $\alpha$ larger than 0.11 is substantial in the MYJ simulation. In the MYJ simulation 43.6 % (22.3 %) of the values in the eyewall region (outer cyclone region) are larger than 0.11. This portion is even larger for the right back quadrant of the MYJ simulation, where the median is larger than 0.11 and 53.3 % are larger than 0.11.*

(e) *The analysis of the shear and veer distribution is sensitive to the definition of the eyewall and outer cyclone region. At the inner edge of the eyewall and along the rainbands larger $\alpha$ and veer values are simulated.*

(f) *Veer and $\alpha$ are larger close to the surface. In particular, wind veer is up to 18 % larger at the rotor bottom than at hub height in the MYJ simulation.*

23. Line 312: This conclusion is drawn based on median wind characteristics. What about wind characteristics that are near the tail of the distribution (e.g., 0.75 percentile)?
We now comment on the fraction of $\alpha$ lager than 0.11 as follows: *The fraction of profiles with $\alpha$ larger than 0.11 is substantial in the MYJ simulation. In the MYJ simulation 43.6 % (22.3 %) of the values in the eyewall region (outer cyclone region) are larger than 0.11. This portion is even larger for the right back quadrant of the MYJ simulation, where the median is larger than 0.11 and 53.3 % are larger than 0.11.*
**Old:** *This supports the conclusion, that the current IEC standard is sufficiently similar in terms of wind shear during tropical cyclones.*

24. Figure 7, Figure 8, and Table 2: The wind profiles representing the 0.75 percentile in Figure 7 display much larger shear than the median wind profiles. This is also evident in the inter quartile range for $\alpha$ reported in Table 2 and in the distribution of $\alpha$ in Figure 8. Please comment on the percentage of wind profiles with shear exponent larger than 0.11. Based on Table 2, it seems about 25% of wind profiles may display shear larger than 0.11. We now include the percentage in the table and comment on the fraction in the method section: *For wind shear, it is of interest if the simulated $\alpha$ is lower than 0.11, which is used in the IEC maximal wind speed model.*

*While the IQR of the $\alpha$ values varies little between the different simulations, the percentage of $\alpha$ values larger than 0.11 depends on the simulation, as well as the specific parts of a tropical cyclone. For the YSU simulation, $\alpha$ is mostly smaller than 0.11 for the analyzed scenes, namely before being affected significantly by land. Only 3.8 % (0.9 %) of the values are larger than 0.11 in the outer cyclone (eyewall) in the YSU simulation. The fraction is larger for the MYNN simulation. MYNN produces 10.4 % of the $\alpha$ values larger than 0.11 in the outer cyclone region. For MYJ with the largest median $\alpha$, a large fraction of $\alpha$ exceeds 0.11. The percentages larger than 0.11 are 43.6 % in the eyewall and 22.3 % in the outer cyclone region.*

25. Figure 8: Why are the y-axis tick labels for shear and veer larger than 1 if this is a plot of probability density? The pdf is defined such that its integral equals one. The probability for a value to be within the interval $x_1$ and $x_2$ is given by Eq. 6. If for example 90 % of the shear values are between 0 and 0.2 the left-hand side of Eq. 6 equals to $P(0 < X \leq 0.2) = 0.9$. In order for the right-hand side of Eq. 6 to be 0.9, the maximum of the pdf needs to be at least $(0.2 - 0.0)/0.9 = 4.5$.

$$P(x_1 < X \leq x_2) = \int_{x_1}^{x_2} \mathrm{pdf}(x)\,dx \tag{6}$$

**2.4 References**

- Kapoor, A., Ouakka, S., Arwade, S. R., Lundquist, J. K., Lackner, M. A., Myers, A. T., Worsnop, R. P., and Bryan, G. H.: Hurricane eyewall winds and structural response of wind turbines, Wind Energ. Sci., 5, 89–104, https://doi.org/10.5194/wes-5-89-2020, 2020.

- Li, X., Pu, Z., and Gao, Z.: Effects of Roll Vortices on the Evolution of Hurricane Harvey During Landfall, Journal of the Atmospheric Sciences, https://doi.org/10.1175/JAS-D-20-0270.1, 2021.

- Ren, H., Dudhia, J., Ke, S., and Li, H.: The basic wind characteristics of idealized hurricanes of different intensity levels, Journal of Wind Engineering and Industrial Aerodynamics, 225, 104980, https://doi.org/10.1016/j.jweia.2022.104980, 2022.

- Ren, Y., Zhang, J. A., Guimond, S. R., and Wang, X.: Hurricane Boundary Layer Height Relative to Storm Motion from GPS Dropsonde Composites, Atmosphere, 10, 339, https://doi.org/10.3390/atmos10060339, 2019.

- Xu, H., Wang, H., and Duan, Y.: An Investigation of the Impact of Different Turbulence Schemes on the Tropical Cyclone Boundary Layer at Turbulent Gray-Zone Resolution, JGR Atmospheres, 126, https://doi.org/10.1029/2021JD035327, 2021.

**3 Reviewer comments from the second referee**

**3.1 General comments**

This manuscript presents a study of examining the sensitivity of low-level wind speed, shear, and veer of typhoon Megi simulated by WRF to three PBL schemes. The goal is to assess how tropical cyclone (TC) force winds affect the wind loading of turbines. The research is interesting and valuable to costal wind engineering. The topic is suitable for Wind Energy Science. The manuscript overall is well written and easy to follow. However, I have a few comments about the manuscript. The manuscript may be published after major revision.
Thank you for the constructive feedback.

**3.2 Specific comments**

1. How PBL schemes affect TC simulations is an old topic. It has been extensively investigated in the past. The three PBL schemes, namely, YSU, MYJ, and MYNN schemes, investigated in this study has been evaluated previously by other studies. For example, Zhu et al. (2013) investigated how these three schemes affect eyewall asymmetric structure and mesovortices of TCs. Recently, Ye et al. (2023) and Ye et al. (2023) also investigated the impact of vertical turbulent mixing parameterization on TC intensification. Although this manuscript focuses

on the issues from the wind engineering perspective different from previous studies, it is important for the authors to do a thorough literature review on this topic, so that the readers know the background of this research and what's new of this manuscript.

- Zhu, P., K. Menelaou, Z.-D. Zhu, 2013: Impact of sub-grid scale vertical turbulent mixing on eyewall asymmetric structures and mesovortices of hurricanes Quart. J. Roy. Meteor. Soc.,140, 416-438. DOI:10.1002/qj.2147.
- Ye, L., Li Y. B., Zhu P., and Gao Z. Q., 2023: The effects of boundary layer vertical turbulent diffusivity on the tropical cyclone intensity, Atmos. Res., 295, 13pp, https://doi.org/10.1016/j.atmosres.2023.106994.
- Ye, G., Zhang X., and H. Yu, 2023: Modifications to Three-Dimensional Turbulence Parameterization for Tropical Cyclone Simulation at Convection Permitting Resolution, J. Adv. Mod. Earth Sys., 15(4), https://doi.org/10.1029/2022MS003530.

Thank you for the additional literature. We have now included them in the new version. We are aware of the large body of existing literature addressing the influence of SGS turbulent fluxes on tropical cyclones. The following two paragraphs have been extended with the suggested literature.

**New:** *The impact of SGS turbulent fluxes on tropical cyclone simulations has been widely investigated. Ye et al. (2023a) show how the spatial distribution of SGS turbulent fluxes depends on the boundary layer closure and how the SGS fluxes affect the tropical cyclone wind field. It has been shown, that the choice of the boundary layer scheme affects the tropical cyclone intensity (Gopalakrishnan et al., 2013; Rai and Pattnaik, 2018; Rajeswari et al., 2020; Zhang et al., 2020; Ye et al., 2023b), the storm radius (Gopalakrishnan et al., 2013; Ye et al., 2023b), the boundary layer inflow strength (Gopalakrishnan et al., 2013; Rajeswari et al., 2020; Zhang et al., 2020) and the inflow layer depth (Rai and Pattnaik, 2018; Gopalakrishnan et al., 2013; Chen, 2022). This suggests, that for tropical cyclones, the three parameters, namely, mean wind speed, wind shear, and veer can be sensitive to the surface and boundary layer parametrization.*

*Old: It has been shown, that the choice of the boundary layer scheme affects the tropical cyclone intensity (Gopalakrishnan et al., 2013; Rai and Pattnaik, 2018; Rajeswari et al., 2020; Zhang et al., 2020), the storm radius (Gopalakrishnan et al., 2013), the boundary layer inflow strength (Gopalakrishnan et al., 2013; Rajeswari et al., 2020; Zhang et al., 2020) and the inflow layer depth (Rai and Pattnaik, 2018; Gopalakrishnan et al., 2013; Chen, 2022).*

*In Nolan et al. (2009) the boundary layer parametrization parameters are affected if eyewall vorticity maxima are developed in mesoscale simulations. This suggests, that the boundary layer parametrization affects wind speed variability in tropical cyclones.*

2. This manuscript aims to address the impact of TC force winds on the wind loading of turbines. Yet, the simulations and the analyses presented in the manuscript focus on the TC winds on the open ocean. One would wonder, why winds on the open ocean can affect the wind turbines on shore? It is the winds at landfall that matter to the turbines. But TC force winds at the open ocean and landfall can be substantially different because of the complicated surface conditions at the coast (e.g., land-ocean contrast, surface roughness, topography, etc.). In this regard, more justification is needed to explain why the open ocean wind matters to turbines on shore.

Offshore wind turbines are increasingly planned in regions affected by tropical cyclones (4Coffshore, 2023). These turbines are located over the ocean, although mostly in close proximity to land. Clearly, nearby land modifies the tropical cyclone wind field. Yet, most model studies on the tropical cyclone wind structure address tropical cyclones over the open ocean. For wind energy a number of studies focus on idealized LES simulation over water (Worsnop et al., 2017; Kapoor et al., 2020; Ren et al., 2022; Gomez et al., 2023). The comparison with these studies is simplified by focusing on the period where the tropical cyclone is over the open ocean. How wind shear and wind veer change upon landfall will be assessed in further studies. We now include the following in the manuscript: *Even though offshore wind projects are mostly limited to coastal regions, we chose to focus on the cyclone intensification stage over open water before the typhoon makes landfall. This stage is chosen for two reasons: 1.) Temporal and spatial averaging of the wind field is only reasonably applicable in the absence of abrupt surface changes. 2.) The comparison to literature is simplified, as the majority of model studies addressing the tropical cyclone wind structure focus on tropical cyclones over the open ocean. How the wind field over land and in close proximity to land differs from the open ocean should be addressed in further studies, where our study can serve as a baseline.*

*Old: To reduce the complexity related to friction over land, we chose to focus on the cyclone intensification stage over open water, before the typhoon makes landfall.*

3. The similarity and difference among the three PBL schemes, YSU, MYJ, and MYNN, are not well summarized. Vertical turbulent mixing schemes can be grouped into three categories in terms of order of closure, local/non-local mixing, and consideration of dry/moist thermodynamics. YSU is a first-order K-closure "dry" scheme that is formulated using thermodynamic variables (e.g., potential temperature, water vapor mixing ratio) that are not conserved for moist reversible adiabatic processes. But it considers non-local mixing. MYJ is a 1.5-order local Turbulent kinetic energy (TKE) "dry" scheme that is also formulated using thermodynamic variables that are not conserved for moist reversible adiabatic processes. Like MYJ, MYNN is a 1.5-order local TKE scheme, but it is formulated with thermodynamic variables that are conserved for moist reversible adiabatic processes. Therefore, it includes the cloud effects on the TKE buoyancy production.

Thank you for including your summary. We have rewritten the paragraph. We closely follow your recommendation, which is similar to Zhu et al. (2014):

**New:** *The three schemes use different ways to calculate SGS turbulent fluxes. The YSU scheme uses a non-local first-order K-closure. The MYJ and MYNN schemes use a 1.5 order local Turbulent kinetic energy (TKE) closure. MYNN is formulated based on variables conserved for moist reversible adiabatic processes and is therefore often called a "moist" scheme (Zhu et al., 2014). Differently, YSU and MYJ are "dry" schemes. The three schemes are widely used in WRF. Due to its non-local closure, YSU is a popular choice to simulate tropical cyclones. Many studies analyze tropical cyclones simulated with the MYJ scheme (Nolan et al., 2009; Sparks et al., 2019; Rajeswari et al., 2020; Shenoy et al., 2021), partly because it was one of only two boundary layer options in the earlier version of WRF (V2.2). The MYNN scheme is an important option for wind resource assessment in the presence of wind farm effects, because wind turbine parametrizations are available for the scheme (Fitch et al., 2012; Volker et al., 2015).*

**Old:** *The YSU scheme is a first-order non-local scheme. Its eddy viscosity is described by a parabolic profile, which is itself a function of boundary layer height. As a non-local scheme, YSU considers that turbulence can be regarded as surface-driven (Kepert, 2012). More precisely, in the YSU scheme, the turbulent tendencies at each vertical level are related to the heat and temperature profile throughout the boundary layer (Nolan et al., 2009). This scheme is however sensitive to the definition of the boundary layer height (Kepert, 2012), while dynamic and thermodynamic definitions of the boundary layer height lead to different results (Zhang et al., 2011). Most studies find, that YSU performs well for tropical cyclones (Rajeswari et al., 2020). In contrast to the YSU scheme, MYJ and MYNN are local schemes. Their eddy viscosity is defined as a function of the turbulent kinetic energy, which is calculated independently at all model height levels. Many studies analyze tropical cyclones simulated with the MYJ scheme (Nolan et al., 2009; Sparks et al., 2019; Rajeswari et al., 2020; Shenoy et al., 2021), partly because it was one of only two boundary layer options in the earlier version of WRF (V2.2). The MYNN scheme is an important option for wind resource assessment in the presence of wind farm effects because wind turbine parametrizations are available for the scheme (Fitch et al., 2012; Volker et al., 2015).*

4. To evaluate the impact of TC winds on turbine wind loading, the authors divided the TC winds into two regimes, eyewall and rainband based on the radii to the storm center. It is unclear why the eyewall region is defined from 60 to 120 km in radius and rainband region is defined from 200 to 400 km. In fact, such definitions can cause problems in wind analyses performed in this study. As shown in Figure 6, the winds at the inner edge of the defined eyewall are much weaker ( 15 – 25 m/s) than those at the outer edge of the eyewall. The former is in the same range as that of the rainband region. But according to the authors, the eyewall region is considered to be a high wind regime, then, what is purpose to include these low winds in a high wind regime? It does not address the problem raised in the paper but only creates a large spread for the eyewall regime (Fig. 7). The definition of rainband regime also has problems. As shown in Figures 5d, 5e, and 5f, the width of rainbands is actually very narrow. In-between the eyewall and rainbands are the non-convective moat, which has different characteristics from the convective eyewall and rainbands. The current definition of rainband region mixes up the convective rainbands and non-convective moat. Such mix-up will have a significant impact on the wind shear and veer analyses as it clearly shows in Figures 5e and 5f that the wind shear and veer in the convective rainbands are substantially different from those in the moat.

Here are my suggestions. Since the authors are interested in the winds in the lower part of the boundary layer, it's better to classify the TC winds into three regimes: eyewall, moat, and rainbands. For the eyewall region, the authors may consider defining it based on the maximum wind speed and the associated radius of the maximum wind (RMW), say, [RMW-R1, RMW+R2] where winds are greater than 80% of maximum wind speed. As for the rainband region, the authors may consider defining it based on either vertical velocity or hydrometeor mixing ratio, i.e., $|w| > wcrit$ or $qh > qcrit$. The rest area in the TC inner core can be grouped into the moat region.

Thank you for the concrete suggestion. We tried to follow your suggestions. We could closely follow your suggestion for the definition of the eyewall region. However, we used the 80 percentile of the 10 m wind speed

as a critical value. We decided to use the percentile instead of the maximal wind speed because the maximal wind speed fluctuates strongly over time. This would lead to strong changes in the selected eyewall area over time. The percentile is more stable over time, and with that the size selected area.

We tested different criteria for the rainbands. Using the vertical winds or the rainwater content resulted in areas that did not align well with the maximal shear and veer values. We came up with a selection criteria based on convective inhibition instead. However, the analyzed statistics of all variables are sensitive to the definition of the eyewall. This includes two aspects: 1.) The size of the area and 2.) how well the area overlays with the outliers of shear and veer. Both aspects may differ between different simulations. This makes it hard to assess the sensitivity of the boundary layer schemes on wind speed, shear, and veer in the rainband and moat area. Therefore, we decided not to separate the two areas in the analysis. We however comment on the different characteristics of rainbands and non-conductive moat. In order to avoid confusion, we further decided to refer to the region combining moat and rainbands as the "outer cyclone region".

In the manuscript we explain the new selection criteria as follows (please refer to the manuscript for Fig. 2):

*The eyewall and outer cyclone regions are analyzed separately. With that, we can account for and characterize differences between the two storm regions. To avoid, that the position of the simulation domain relative to the cyclone center influences the analysis, we use only grid points within a distance to the cyclone center (R) smaller than 350 km. The definitions of the eyewall and outer region are illustrated in Fig. 2 and summarized in Eq. 4 and 5. We define the eyewall as a high wind speed regime. Our definition is based on the simulated wind speed at 10 m ($WS_{10}$) at each simulation time step. Grid points are assigned to the eyewall region if two criteria are fulfilled: 1.) $WS_{10}$ is greater than or equal to the 80th percentile of $WS_{10}$ ($P_{80}(WS_{10})$), and 2.) $R$ is less than 250 km. The selected eyewall area is 76000 $km^2$. The thickness of the eyewall is not symmetrical over the azimuth and can be zero. Note also, that the area includes gaps, where the wind speed is lower than $P_{80}(WS_{10})$. The outer region includes all grid points that are not part of the eyewall and the eye. To distinguish between the outer region and the eye, we define a critical radius $R_{Eye}$ in Eq. 5. Here, $R_{Eye}$ is taken as the 10th percentile of $R$ of the eyewall grid points ($P_{10}(R_{Eyewall})$). Thus*

$$Region = \begin{cases} Eyewall: & if\ WS_{10} \geq P_{80}(WS_{10})\ and\ R < 250\,km, \\ Outer\ cyclone: & if\ not\ Eyewall\ and\ R > R_{Eye} \end{cases} \tag{7}$$

$$R_{Eye} = P_{10}(R_{Eyewall}) \tag{8}$$

*The outer region is not homogeneous. It includes both the rainbands and the non-convective moat areas. These areas have different properties. However, we do not separate them, to prevent the analysis from becoming sensitive to the selection criteria.*

We further comment on differences along the rainband various times. We include here three passages:

(a) *Yet, along the spiraling rainbands, $\alpha$ increases around $3 \times 10^{-2}$ between the hub height and the rotor top (not shown).*

(b) *In the outer cyclone region, the maximal values of shear and veer are found along the spiraling rainbands (Fig. 6). Along the rainbands, there is a zone of lower horizontal wind speed. Within this zone, local maxima and minima of wind shear are alternating. The wind veer changes from positive values (inflow angle decreasing with height) on the radially inward side of the rainbands to negative values (inflow angle increasing with height) on the outside.*

(c) *At the inner edge of the eyewall and along the rainbands larger $\alpha$ and veer values are simulated.*

5. The authors argue that in addition to the wind speed, wind shear and veer are also important to turbine wind loading. But only the vertical profiles of wind speed in the defined eyewall and rainband regions are shown (Fig. 7). Then, what about the wind shear and veer. In fact, vertical profiles of wind speed in TCs have been shown in many papers, but not the shear and veer. So, it would be interesting to see what the vertical profiles of wind shear and veer in different regimes of a TC look like.

Thank you for the suggestion. We now include profiles of wind veer and wind shear in Fig. 8 and Fig. 9. In the result section we refer to the shear and veer profiles as follows:

[revised manuscript text omitted]

---

## Referee Report (RR1)

2nd Review of manuscript wes-2023-71 entitled "Tropical cyclone low-level wind speed, shear, and veer: sensitivity to the boundary layer parameterization in WRF" by Sara Muller, Xiaoli Guo Larsen, and David Verelst

General comments

Thanks to the authors' great effort, the revised manuscript has been much improved. The authors did consider my comments for the paper revision although they did not follow some of my suggestions, particularly for how to define the eyewall region. But I respect their decisions.

I have two additional comments. First, in the introduction, the authors stated that wind veer should be considered in wind turbine load assessment (line39), but it is not accounted for the IEC standards (line 35). So, one of the objectives of this paper, I think, should be the examination of how wind veer can be included in the IEC standards for wind turbine loading based on this research. But this issue has not been clearly addressed in the paper. Based on the research, a recommendation should be made in the conclusion or discussion whether wind veer should be included in the IEC standards and how.

Second, the eyewall is now defined as the region of $80^{th}$ percentile of 10-m wind speed and the radius is smaller than 250 km. But 250 km is a large radius, which would include both eyewall and inner rainbands. It could also include a part of the outer rainbands. So, such defined "eyewall region" is not really the eyewall commonly referred to in tropical meteorology. This can cause confusion. Here are my two suggestions. If possible, I'd suggest defining the "eyewall region" as the area with 80th percentile of 10-m wind speed and within the vicinity of the maximum wind (RMW), i.e., [RMW-dR, RMW+dR]. This is closer to the traditionally defined eyewall. If this is a burden for the authors as they have to redo all the analyses, then, alternatively, the authors could specifically note in the paper that such defined "eyewall region" should not be interpreted as the true eyewall in tropical cyclones, rather, it refers to a rough region with high wind speeds that includes the eyewall and inner rainbands, or maybe a part of the outer rainbands.

Other comments

1. Although Figure 2 is an illustration figure. It is not appropriate to use longitude and latitude as x- and y-axis without any marker. I'd suggest using radius so that readers can have a rough idea of the size of the storm and ranges of defined eyewall region and outer cyclone.

2. Caption of Figures 8, it reads "Profiles of a,b) wind speed, and shear exponent c,d) for the a,c) eyewall region, and b,d) outer cyclone region". This is an awkward sentence with so many a,b), c,d), a,c), and b,d). Please simplify the sentence. You can just say: "a, c) and b, d) are the vertical profiles of wind speed and shear exponent for the eyewall region and outer cyclone, respectively. The same problem is for Figure 9 caption.

---

## Author Response (AR2)

**Answer to reviewer comments**

March 16, 2024

**1 Summary of changes**

Many thanks for your suggestions. We made the following adjustments:

- Updated citation of Sanchez Gomez et al. (2023).

- Refrasing the introduction of addressing the importance of wind veer.

- Comments on eyewall definition in the Method section.

- Quantification of sensitivity to eyewall definition in Discussion.

- Refracing labels for Figures 2, 8, and 9.

- Change of axis labels in Figure 2.

Detailed reasoning for the changes is given in Sect. 2.2.

**2 Resoponse to comments**

**2.1 General Comments**

1. Both reviewers agree that substantial revisions have greatly improved the manuscript.
   Thank you, we are glad that the revisions are well received.

2. Reviewer 2 has a few suggestions that should be easily handled, included below, including an important distinction regarding the definition of the eyewall.
   The comments from reviewer 2 are answered in Sect. 2.2.

3. The editor notes that references to "Gomez et al." (lines 40, 410-413, 577) should be references to "Sanchez Gomez et al." rather than "Gomez et al.".
   The reference is corrected.

**2.2 Comments from Reviewer 2**

I have two additional comments.

1. First, in the introduction, the authors stated that wind veer should be considered in wind turbine load assessment (line39), but it is not accounted for the IEC standards (line 35). So, one of the objectives of this paper, I think, should be the examination of how wind veer can be included in the IEC standards for wind turbine loading based on this research. But this issue has not been clearly addressed in the paper. Based on the research, a recommendation should be made in the conclusion or discussion whether wind veer should be included in the IEC standards and how.
   Thank you for this comment! Indeed we don't make a clear statement on the inclusion of veer in the IEC. We are cautious about making a strong recommendation on whether veer needs to be included in the IEC standard. Based on our study we can state, that the veer over the open ocean in mesoscale simulations is relatively small compared to veer in stably stratified low-wind regimes. Yet, 1.) we did not assess the impact of wind veer on wind turbine loads under tropical cyclone conditions, and 2.) we had to limit the scope of the study to a case study over the open ocean. Both shortcomings are stated and suggested for further work. Further work is needed to conclude if the contribution of wind veer to turbine loads in tropical cyclone conditions is sufficiently significant to be included in standards. We already address the inclusion of veer in the standards:

- In the discussion: *"Further, the simulated wind veer is relatively small in comparison with wind veer found in low-wind regimes and particularly during stable conditions. This supports the conclusion, that the current IEC standard may be sufficiently similar in terms of wind shear and veer during tropical cyclones over open water. However, there are clear limitations to this conclusion as discussed in the following ..."*

- In the conclusion: *"Based on these conclusions, further investigation is needed to address 1) how wind speed, shear, and veer in tropical cyclones evolve during landfall, 2) how much wind shear and wind veer vary between tropical cyclones with different intensities and radii, and 3) how much wind turbine load estimates based on the current IEC standard differ between load estimates based on the simulated wind speed, shear, and veer distributions."*

You are right, the introduction directly led to the question of whether veer should be included in the standards and provokes the expectation of a clear statement. For that reason, we decided on subtle changes in the introduction. Concretely, we refrain from starting the introduction of wind veer with: "Wind veer, the change in wind direction with height is not accounted for in the IEC standards."

2. Second, the eyewall is now defined as the region of 80th percentile of 10-m wind speed and the radius is smaller than 250 km. But 250 km is a large radius, which would include both eyewall and inner rainbands. It could also include a part of the outer rainbands. So, such defined "eyewall region" is not really the eyewall commonly referred to in tropical meteorology. This can cause confusion. Here are my two suggestions. If possible, I'd suggest defining the "eyewall region" as the area with 80th percentile of 10-m wind speed and within the vicinity of the maximum wind (RMW), i.e., [RMW-dR, RMW+dR]. This is closer to the traditionally defined eyewall. If this is a burden for the authors as they have to redo all the analyses, then, alternatively, the authors could specifically note in the paper that such defined "eyewall region" should not be interpreted as the true eyewall in tropical cyclones, rather, it refers to a rough region with high wind speeds that includes the eyewall and inner rainbands, or maybe a part of the outer rainbands.

This is a valid point, thank you for the concrete suggestion. We are aware of the effect of using different definitions and thus, we follow your suggestions and add:

- In the methods: *"This definition of the eyewall region includes high wind speed areas of the inner rainbands and potentially outer rainbands. Therefore the eyewall region should not be interpreted as the narrow eyewall in tropical cyclones, but rather as an extended high wind speed area."*

- In the discussion: *"When restricting the definition of the eyewall region to a narrower band with a width of $0.4 \times$ RMW, the median wind speed in the eyewall increases on the order of 12 %, the shear exponent 4 %, and veer 10 %."*

For the second statement, we tested an alternative definition of the eyewall region. Here we provide some further information, which is not included in the paper. We added two criteria for the alternative definition of the eyewall region: 1) the mean eyewall width is $0.4 \times$ the RMW (as used for distributions of shear and veer in Figure 8 of Sanchez Gomez et al. (2023)), and 2) the eyewall extends maximal to $1.5 \times$ RMW. The resulting distributions are summarized in Table 1 (not included in the manuscript).

| Region | Scheme | Wind speed [m s$^{-1}$] median | IQR | Shear exponent median | IQR | % > 0.11 | Wind veer [° m$^{-1}$] median | IQR |
|---|---|---|---|---|---|---|---|---|
| | YSU | 42.9 | 5.1 | $8.8 \times 10^{-2}$ | $1.0 \times 10^{-2}$ | 1.5 | $1.6 \times 10^{-2}$ | $8.1 \times 10^{-3}$ |
| Eyewall | MYNN | 37.5 | 3.7 | $9.8 \times 10^{-2}$ | $1.0 \times 10^{-2}$ | 7.8 | $1.8 \times 10^{-2}$ | $7.0 \times 10^{-3}$ |
| | MYJ | 45.3 | 6.2 | $1.1 \times 10^{-1}$ | $1.2 \times 10^{-2}$ | 60.1 | $1.8 \times 10^{-2}$ | $7.7 \times 10^{-3}$ |
| | YSU | 27.6 | 7.2 | $7.1 \times 10^{-2}$ | $1.6 \times 10^{-2}$ | 3.3 | $9.5 \times 10^{-3}$ | $7.9 \times 10^{-3}$ |
| Outer cyclone | MYNN | 25.4 | 6.4 | $9.0 \times 10^{-2}$ | $1.8 \times 10^{-2}$ | 9.7 | $1.2 \times 10^{-2}$ | $7.6 \times 10^{-3}$ |
| | MYJ | 27.1 | 7.8 | $10.0 \times 10^{-2}$ | $1.9 \times 10^{-2}$ | 23.4 | $1.2 \times 10^{-2}$ | $8.6 \times 10^{-3}$ |

Table 1: As table 2 in the article but for a eyewall with an average radius of 0.4 RMW: Median and interquartile range (IQR) of wind speed at 139 m, wind shear exponent, and wind veer, as well as the percentage of shear exponent values larger than 0.11. The values are listed for the eyewall region and outer cyclone region for the YSU, MYJ, and MYNN simulation.

**2.3   Other comments**

1. Although Figure 2 is an illustration figure. It is not appropriate to use longitude and latitude as x- and y-axis without any marker. I'd suggest using radius so that readers can have a rough idea of the size of the storm and ranges of defined eyewall region and outer cyclone.
   Good point, we changed the axis of Figure 2 to show the distance from the cyclone center.

2. Caption of Figures 8, it reads "Profiles of a,b) wind speed, and shear exponent c,d) for the a,c) eyewall region, and b,d) outer cyclone region". This is an awkward sentence with so many a,b), c,d), a,c), and b,d). Please simplify the sentence. You can just say: "a, c) and b, d) are the vertical profiles of wind speed and shear exponent for the eyewall region and outer cyclone, respectively.
   Thank you we followed your suggestion.

3. The same problem is for Figure 9 caption.
   The caption to Figure is changed similarly to Figure 8.

**References**

Sanchez Gomez, M., Lundquist, J. K., Deskons, G., Arwade, S. R., Myers, A. T., and Hajjar, J. F.: Wind conditions in category 1-3 tropical cyclones can exceed wind turbine design standards, ESS Open Archive, https://doi.org/10.22541/essoar.168394766.67483870/v1, 2023.